# The role of the human hippocampus in decision-making under uncertainty

Bahaaeddin Attaallah ®[1] ✉, Pierre Petitet ®[2], Rhea Zambellas[1], Sofia Toniolo[1], Maria Raquel Maio[1], Akke Ganse-Dumrath ®[1,2], Sarosh R. Irani[1], Sanjay G. Manohar ®[1,2] & Masud Husain ®[1,2]

The role of the hippocampus in decision-making is beginning to be more understood. Because of its prospective and inferential functions, we hypothesized that it might be required specifically when decisions involve the evaluation of uncertain values. A group of individuals with autoimmune limbic encephalitis—a condition known to focally affect the hippocampus—were tested on how they evaluate reward against uncertainty compared to reward against another key attribute: physical effort. Across four experiments requiring participants to make trade-offs between reward, uncertainty and effort, patients with acute limbic encephalitis demonstrated blunted sensitivity to reward and effort whenever uncertainty was considered, despite demonstrating intact uncertainty sensitivity. By contrast, the valuation of these two attributes (reward and effort) was intact on uncertainty-free tasks. Reduced sensitivity to changes in reward under uncertainty correlated with the severity of hippocampal damage. Together, these findings provide evidence for a context-sensitive role of the hippocampus in value-based decision-making, apparent specifically under conditions of uncertainty.

Humans often face situations where they have to decide whether the reward they might obtain from their actions is worth the cost required for it—for example, when having to allocate effort to accomplish something. Whether it is buying an item in a grocery store or making life-changing resolutions, such trade-offs can influence our decisions and behaviour in our daily lives. Emerging evidence from animal studies suggests that the hippocampus might contribute to reward processing and valuation, with reports indicating several forms of reward representation in the hippocampal formation and its extended networks (for a review, see ref. 1). Theories and empirical reports investigating this possible hippocampal role in humans have tied it to this structure's well-known functions in memory, associative inference and imagination[2,3]. Mechanistically, these investigations implicate the hippocampus in several processes, including the spreading of values between different contexts[4–6], the construction of values from prior experiences[7], updating[8,9] and the stabilization of preferences[10].

One evolving concept connecting these unique properties of the hippocampus proposes that it provides context against which reward is evaluated to support value-based decisions and preferences[11,12]. This process might be mediated by hippocampus-dependent mental time travel into both the past (sampling from memory) and the future (sampling from projected possible futures) to allocate these contexts. For example, 'preplay' signals (corresponding to reward delivery in yet-to-be-explored environments) and 'look ahead' signals (representing future trajectories leading to goals) have both been recorded in rodent hippocampus[13–15]. Similarly, in humans, functional MRI hippocampal activity has been observed when people make decisions that involve reward anticipation and future considerations[16–18].

With this perspective, the hippocampus might be implicated in the evaluation of reward when episodic thinking is critically involved (for example, to process values of projected possible futures)[19]. Such scenarios involve probabilistic consideration of future value states

[1]Nuffield Department of Clinical Neurosciences, University of Oxford, Oxford, UK. [2]Department of Experimental Psychology, University of Oxford, Oxford, UK. ✉e-mail: Bahaaeddin.Attaallah@nhs.net

(that is, making decisions under uncertainty)[20,21]. By contrast, this conceptualization of the hippocampal role in motivated behaviour suggests that contexts of a deterministic nature (for example, when evaluating rewards against known physical effort costs) should be less influenced by hippocampus-related prospective computations.

In a recent report, we demonstrated that the hippocampus might be implicated in active information gathering prior to committing to decisions under uncertainty in people with subjective cognitive impairment[22]. Markers of increased reactivity to uncertainty (for example, rapid collection of information) were found to be associated with heightened hippocampal–insular connectivity. This finding aligns in part with previous studies highlighting hippocampal contribution to uncertainty processing and related forms of decision-making such as inter-temporal choices and visual information search[23–27]. However, it remains unclear whether this proposed role of the hippocampus in valuation and decision-making is contextually specific to uncertainty (that is, implicated only when agents have to consider uncertainty) or reflects a general hippocampal processing of reward and value regardless of contexts.

To answer this question and to directly investigate hippocampal involvement in goal-directed decision-making and reward valuation, we recruited 19 people with autoimmune limbic encephalitis (ALE)—a rare neurological condition known to affect the hippocampus. Patients in the chronic phase of ALE characteristically have highly focal hippocampal atrophy[28–35], making them an ideal model of selective hippocampal dysfunction that is well suited to making inferences on the basis of structure–function correlations[35–38]. This is especially feasible in such an experimental group, as the extent of hippocampal damage varies between ALE patients depending on the course of the illness and the interval between disease onset and treatment initiation[29,33].

The participants (patients and healthy matched controls) were tested in four experiments examining how people make decisions considering reward, uncertainty and/or physical effort attributes. In the first two experiments, we used the Circle Quest behavioural paradigm, which has been previously tested and validated in healthy people and patients with subjective cognitive impairment[22,39]. The original paradigm has two versions: active and passive (Exps. 1 and 2). The active version of the task examines how people give up rewards to obtain information and reduce uncertainty before committing to decisions. The passive version allows limited agency over uncertainty to examine how people make passive decisions on whether to accept or reject offers on the basis of predetermined levels of reward and uncertainty. In Exp. 3, effort-based decision-making was examined using a modified version of an extensively validated behavioural paradigm used in previous studies in healthy people and individuals with neurological disorders[40,41]. This paradigm has a similar design to the passive version of Circle Quest and examines how people make passive decisions weighing rewards against physical effort. In Exp. 4, a third version of Circle Quest was introduced to investigate how people make passive decisions considering the three attributes of interest (reward, uncertainty and physical effort). In other words, participants in Exp. 4 were required to make decisions weighing the reward on offer given both the physical effort cost and the uncertainty in the environment.

The results from these experiments converged to indicate that in ALE patients, despite intact sensitivity to uncertainty, the presence of uncertainty is associated with blunted sensitivity to other value attributes (reward and effort cost). In the active version of the Circle Quest task (Exp. 1), patients were less sensitive to the cost of sampling when gathering information to support decisions under uncertain conditions, resulting in faster, extensive and wasteful sampling when the cost of sampling and the reward on offer increased. In the passive version of Circle Quest (Exp. 2), ALE patients were significantly less sensitive to changes in reward, and this effect correlated with lower sensitivity to sampling cost changes observed in active sampling (Exp. 1) as well as with the severity of hippocampal atrophy. When assessed on the effort-based decision-making task (Exp. 3), no significant difference was found between patients and controls when they made effort-based decisions without uncertainty, indicating intact valuation of reward and effort under conditions that do not feature uncertainty. By contrast, on the third version of Circle Quest (Exp. 4), patients were less sensitive to changes in effort and reward than controls, while their sensitivity to uncertainty was intact. Intact sensitivity to uncertainty was observed across all versions of the Circle Quest paradigm (Exps. 1, 2 and 4).

Taken together, these results indicate an uncertainty-sensitive role of the hippocampus in value-based decision-making. These findings might represent an important additional step in understanding selective hippocampal contributions to goal-directed and motivated behaviour.

## Results

### Experimental paradigms

**Exp. 1—active information gathering prior to committing to decisions under uncertainty.** In the first experiment (Exp. 1), participants completed a shorter version of the Circle Quest paradigm, a recently developed behavioural task investigating active information sampling before committing to decisions under uncertainty[22,39]. The participants were asked to maximize their reward by localizing a fixed-size hidden circle as precisely as possible. Uncertainty about the precise location of the hidden circle could be reduced by gathering information through touching the screen at different locations. If the location where they touched was situated inside the hidden circle, a purple dot appeared; and if the location was outside that hidden circle, a white dot appeared. The participants started each trial with an initial credit reserve ($R_0$) from which the cost to obtain a new sample ($\eta_s$) was subtracted with each additional sample. After the active sampling phase, during which the participants could sample the screen for information without restriction of speed or location, a blue disk matching the size of the hidden circle appeared. The participants were then required to move this blue disk to where they thought the hidden circle was located. Depending on localization error (how far the blue disk centre was from the true location of the hidden circle) and the cost of sampling, the score for each trial was calculated and provided as feedback at the end of the trial (equation in Fig. 1). The task thus imposed an economic trade-off between the benefit and cost of obtaining information. There were two levels of sampling cost (low and high) and two levels of initial credit reserve (low and high). Uncertainty—indexed by circle localization expected error (EE)—was quantified as the probability-weighted average of all the possible errors that could occur upon placing the localization disk (Fig. 1d and Methods). To expose the participants to the task environment and its scoring, the testing session began with a training task in which they practised circle localization at various levels of uncertainty and reward. This also helped establish the effect of visuospatial demand on localization performance.

**Exp. 2—passive decision-making under uncertainty.** In the second experiment (Exp. 2), the participants performed a second version of the Circle Quest paradigm examining how they weighed potential rewards against uncertainty when making decisions. Eight dots (four purple and four white) were always presented on the screen in each trial. The spatial configurations of these dots were manipulated experimentally to produce different levels of uncertainty—for example, when the purple dots were spaced widely apart, the location of the hidden circle was less uncertain than when they were clumped closer together because the former configuration imposes more limitations on possible circle placements. To limit memory load, a circle of the same size as the hidden circle was always present on each side of the screen to provide a continuous reminder of its size.

On each trial of the passive task, the participants were asked to report uncertainty estimates (confidence ratings on a scale from

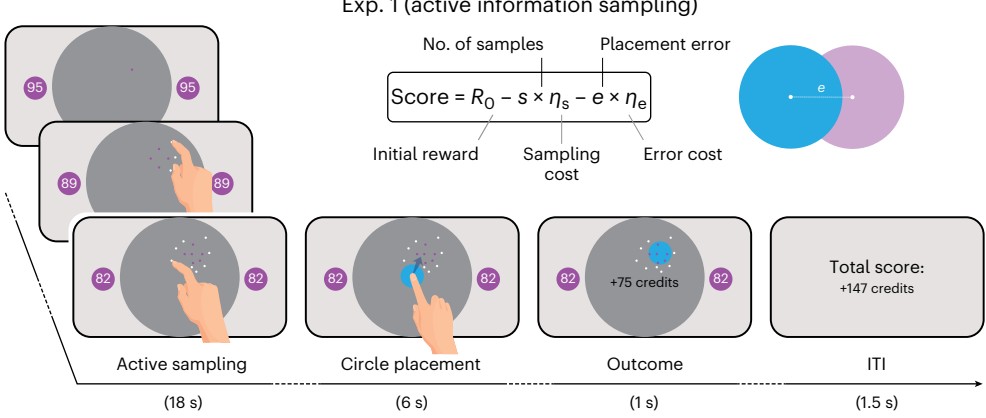

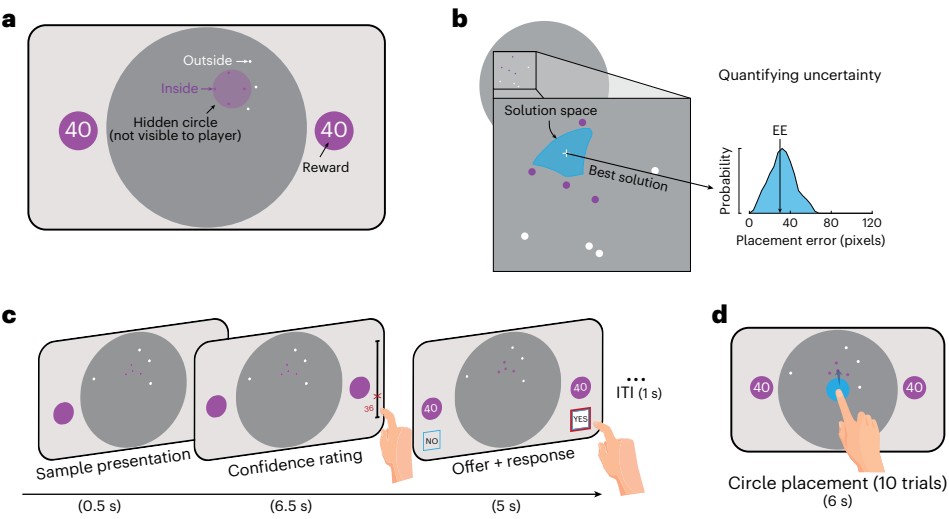

**Fig. 1 | Task paradigms for Exps. 1 and 2.** Exp. 1 investigated active information gathering. Each trial began with a purple dot giving an initial hint about the hidden circle's location. The participants touched the screen to reveal further clues (purple or white dots), narrowing down the solution space of the location of the hidden circle. Purple dots indicated a location inside the hidden circle, whereas white dots signalled outside locations. The participants were free to touch anywhere on the screen within the 18-second trial period, with the option to stop anytime. However, each touch providing additional information reduced the initial reward reserve $R_0$, shown inside the two purple circles on the side of the screen. For example, in the illustrated trial, the participants started with 95 credits and lost one credit for each additional sample. After the active sampling phase (18 seconds), a blue disk appeared automatically at the screen's centre, which the participants then moved to their estimated hidden circle location. ITI, intertrial interval. Trial scores were calculated by subtracting a localization error penalty ($e \times \eta_e$, where $e$ is the error in pixels and $\eta_e$ the error

cost in credits per pixel) from the remaining reward reserve ($R_0 - s \times \eta_s$, where $s$ is the number of samples and $\eta_s$ the sampling cost). The error cost $\eta_e$ was constant and equal to 1.2 credits per pixel. Exp. 2 investigated passive decision-making under uncertainty. The task examined how people weigh potential rewards against uncertainty when making decisions. **a**, An example of an offer with a search configuration representing uncertainty and credits indicating the reward on offer. **b**, Different spatial configurations were associated with different uncertainty levels quantified as EE, which is equal to the probability-weighted average of all the possible errors that could result when placing the localization disk at the best possible location. **c**, The participants first reported their subjective estimation of uncertainty (their confidence about the circle location). After this, the credits on offer appeared, and the participants decided whether to accept or reject the offer. **d**, At the end of the experiment, ten of the accepted offers were played (placing the blue disk), determining the final score. Hand icon from macrovector on Freepik.

0 to 100) reflecting how well they thought they might be able to locate the hidden circle, given the configuration of dots on the screen. Next, they were presented with the reward on offer and were asked, 'Do you want to play this trial for this potential reward?' to which they could respond either 'Yes' or 'No' (by pressing the corresponding answer on the touchscreen). The participants were told that 10 of their 'Yes' responses from the 100 trials played would be randomly selected at the end of the experiment, and that they would have to place a blue disk (of the same size as the hidden circle) where they thought the hidden circle was located for each of these trials. Their monetary reward would be based on their localization performance on these 10 trials. If they located the hidden circle perfectly (that is, if they placed the blue disk exactly on top of the hidden circle), they won all credits on

offer. If not, they lost credits proportionally to the magnitude of their localization error.

**Exp. 3—effort-based decision-making.** To investigate effort-based decision-making, we used a modified version of a well-validated effort-based decision-making task that has been extensively used in healthy people and different patient groups[41–46]. This task had a similar design as the passive choices task used in Exp. 2 to investigate reward valuation against uncertainty (Fig. 1a). Reward was represented as apples on trees that the participants were asked to weigh against physical effort levels that they needed to exert to obtain the apples (Fig. 2). Physical effort in the task pertained to squeezing a hand-held dynamometer up to various force levels. There were five

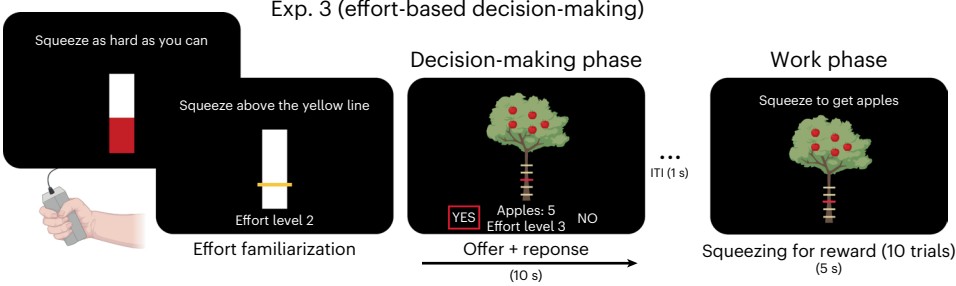

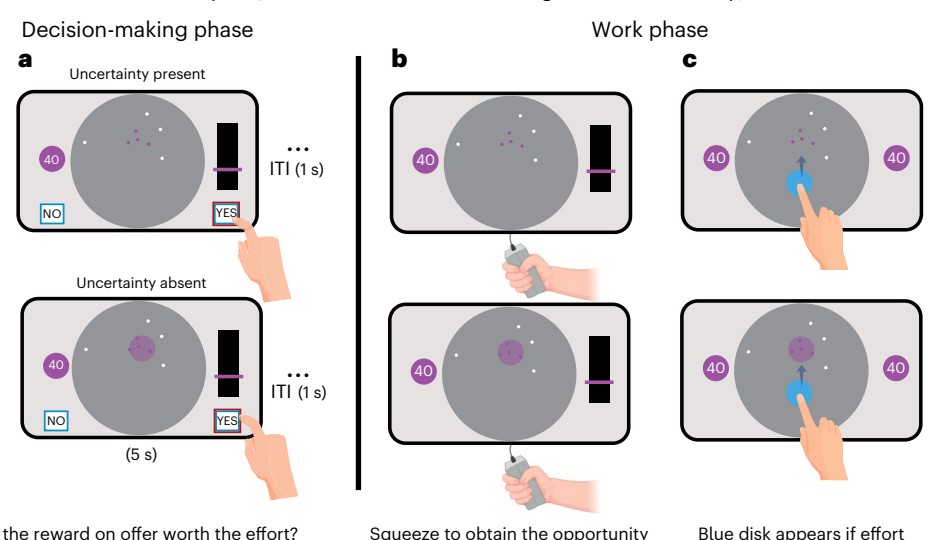

**Fig. 2 | Task paradigms for Exps. 3 and 4.** Exp. 3 investigated effort-based decision-making. The design is similar to that of Exp. 2. Initial calibration involved setting the hand-held dynamometer on the basis of each participant's MVC. The participants then familiarized themselves with various effort levels, squeezing the handle to match the effort indicated by a yellow line; a higher line denoted more effort. They practised this twice per effort level. The core task involved deciding whether the reward (apples) on offer was worth the effort assigned to it. The participants were informed that ten of their choices would be randomly selected at the experiment's end to physically execute for obtaining apples. Exp. 4 investigated effort-based decision-making under uncertainty. ITI, intertrial interval. **a**, Post-training (see below), the participants underwent 200 trials of accept/reject decisions, balancing rewards (credits) against effort levels. Decisions were made considering the presence or absence of uncertainty, particularly in estimating the hidden circle's location on the basis of dot configurations (as detailed in Methods). The absence of uncertainty was signified by directly showing the purple circle's location. **b,c**, After the experiment, 24 trials were randomly chosen from the accepted decisions. Here, the participants exerted physical effort (**b**) to earn the chance to place a blue disk (**c**) where they thought the hidden circle was. In 12 trials, the purple circle's location was shown, and in the other 12, it was not. Performance accuracy determined the final credits earned. Training occurred in three stages: (1) an interactive tutorial on Circle Quest (as in Exps. 1 and 2) introduced the scoring function and assessed localization accuracy (Fig. 8a); (2) the participants rated their confidence in locating the hidden circle on a scale of 0 to 100, with dot configurations reflecting those in the decision-making phase and additional catch trials to broaden the uncertainty range (subjective uncertainty estimates in Fig. 8b); and (3) effort calibration and familiarization mirrored Exp. 3. Hand icons from BioRender.com and macrovector on Freepik.

effort levels corresponding to 16%, 32%, 48%, 64% and 80% of each participant's maximal voluntary contraction (MVC) measured at the beginning of the task. After a familiarization period with these effort levels, the participants made accept/reject decisions for various reward–effort offer combinations. Similar to the passive version of the Circle Quest task (Exp. 2), individuals were instructed that some of the offers they accepted (10 of 125 trials played) would be randomly selected at the end of the experiment to be carried out, and that they would receive monetary rewards based on the number of apples they collected.

**Exp. 4—effort-based decision-making under uncertainty.** In the fourth experiment (Exp. 4), the participants were re-invited to perform a third version of the Circle Quest paradigm, this time designed to investigate effort-based decision-making under uncertainty. The task was similar to those in Exps. 2 and 3. However, instead of making decisions (accept/reject) on the basis of two attributes (reward versus uncertainty as in Exp. 2 and reward versus effort as in Exp. 3), the participants now

had to make decisions under the three attributes together (reward, uncertainty and effort simultaneously) (Fig. 2a).

Before engaging in the decision-making phase, the participants were familiarized with the task using an interactive tutorial and then trained on circle localization (similar to Exps. 1 and 2) and effort practice (similar to Exp. 3). Unlike Exp. 2, where participants estimated uncertainty prior to each decision, participants in this task gave these reports in one block before the decision phase. This accommodated two uncertainty conditions, present or absent, ensuring an experimentally balanced design for decision trials across these two conditions.

On each trial in the decision phase, the participants responded to offers considering three attributes:

- Reward, depicted as credits with the same four levels as Exp. 2.
- Effort, represented by the bar height on a rectangle, mirroring the five levels from Exp. 3. Successfully achieving the effort level unveiled the blue disk used in localizing the hidden circle to win credits.

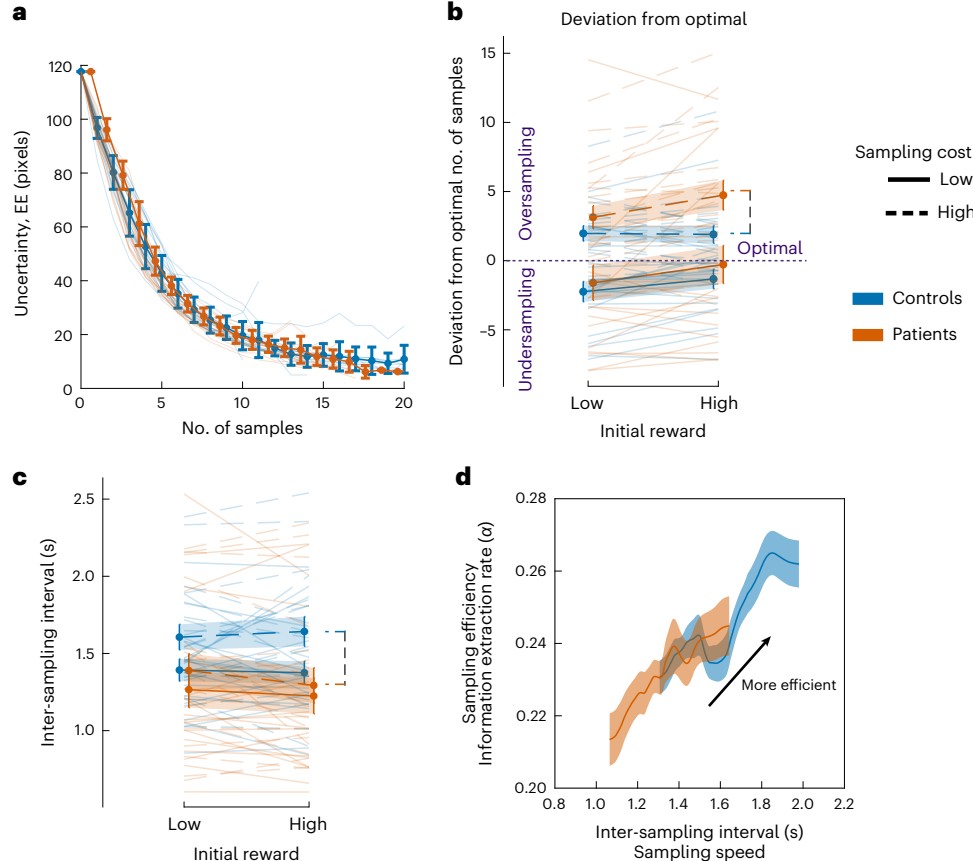

**Fig. 3 | Exp. 1—reduced sensitivity to changes in information cost in ALE patients. a**, Uncertainty (indexed as EE) decreases with sampling and follows an exponential decay slope on average in both patients and controls. **b**, ALE patients and healthy controls oversampled when the sampling cost was high. Patients, however, oversampled to a greater extent than controls, mainly when the initial reward reserve and sampling cost both increased ($z = 2.267$, $P = 0.023$, Cliff's $\delta = 0.43$). There was no significant difference between the two groups under low-cost conditions. **c**, Higher sampling cost was associated with slower sampling rates. This effect was less evident in ALE patients than in controls, resulting in

faster sampling in ALE patients in high-sampling-cost conditions (ALE × $\eta_s$: $\beta = -0.0719$; $t_{2272} = -2.14$; $P = 0.033$; 95% CI, (−0.13, −0.005)). **d**, Sampling behaviour was characterized by a speed–efficiency trade-off whereby faster sampling rates (shorter inter-sampling intervals) were associated with lower sampling efficiency (smaller $\alpha$). The figure shows this trade-off for the high-sampling-cost conditions, in which ALE patients sampled faster and also oversampled when the initial reward was high. The error bars and shading show ±s.e.m. The data are from 19 patients and 19 controls. See Supplementary Tables 4–6 for the full statistical details.

- Uncertainty, showcased through dot configurations (as in Exps. 1 and 2). Uncertainty was absent on half of the trials by revealing the true circle location. When present, uncertainty ranged between EEs of 31.8 and 73.95 pixels, similar to Exp. 2.

Thus, similar to Exp. 2, each trial presented the participants with credit offers attainable upon accurately localizing the hidden circle. However, achieving the effort level designated for the trial was essential to reveal the localization disk. The participants had to make 'Yes/No' decisions across 200 trials, encompassing three economic attributes: uncertainty (present or absent), reward (four levels) and effort (five levels). Ten additional catch trials were included when uncertainty was present, expanding the range of uncertainty beyond that used in the main task. The participants were informed that 24 'Yes' decisions would be randomly selected at the task's end, providing them an opportunity to play and receive rewards.

### Demographics
Demographics and group characteristics are summarized in Supplementary Tables 1 and 2. ALE patients had lower cognitive scores on Addenbrooke's Cognitive Examination III (ACE III) (controls: $\mu = 97.52$, s.d. = 2.03; ALE: $\mu = 93.42$, s.d. = 5.64; $t_{36} = 2.99$; $P = 0.005$; 95% confidence interval (CI), (1.31, 6.89); Cohen's $d = 0.94$). These cognitive differences were seen mainly in two domains of ACE III: memory (controls: $\mu = 24.79$,

s.d. = 1.36; ALE: $\mu = 23.16$, s.d. = 2.54; $z = 2.19$, $P = 0.028$, Cliff's $\delta = 0.40$) and fluency (controls: $\mu = 13.32$, s.d. = 1.36; ALE: $\mu = 11.21$, s.d. = 2.69; $z = 2.81$, $P = 0.004$, Cliff's $\delta = 0.51$). The other three domains including language, visuospatial abilities and attention were not significantly different between the two groups. There was no significant difference in measures of executive function (digit span), apathy (Apathy Motivation Index), fatigue (Fatigue Severity Scale), depression (Beck Depression Inventory-II) or hedonic experience (Snaith–Hamilton Pleasure Scale).

### Exp. 1—reduced sensitivity to changes in information cost in ALE patients
In the active sampling task (Exp. 1), participants in both groups acquired samples to reduce uncertainty. Similar to previous reports[22,39], reduction in uncertainty followed an exponential decay as a function of the number of samples acquired, indicating purposeful sampling abiding by task rules (Fig. 3a). Both patients and healthy controls behaved rationally, sampling less when acquiring samples was more expensive (main effect of $\eta_s$ on the number of samples acquired: $\beta = -0.11$; $t_{2272} = -6.25$; $P < 0.0001$; 95% CI, (−0.14, −0.07); Supplementary Table 4) and not responding to changes in the initial reward reserve (main effect of $R_0$ on the number of samples acquired: $\beta = 0.023$; $t_{2272} = 1.29$; $P = 0.20$; 95% CI, (−0.01, 0.05)).

Patients' and controls' performance in the active task was evaluated with regard to optimal sampling behaviour to determine whether

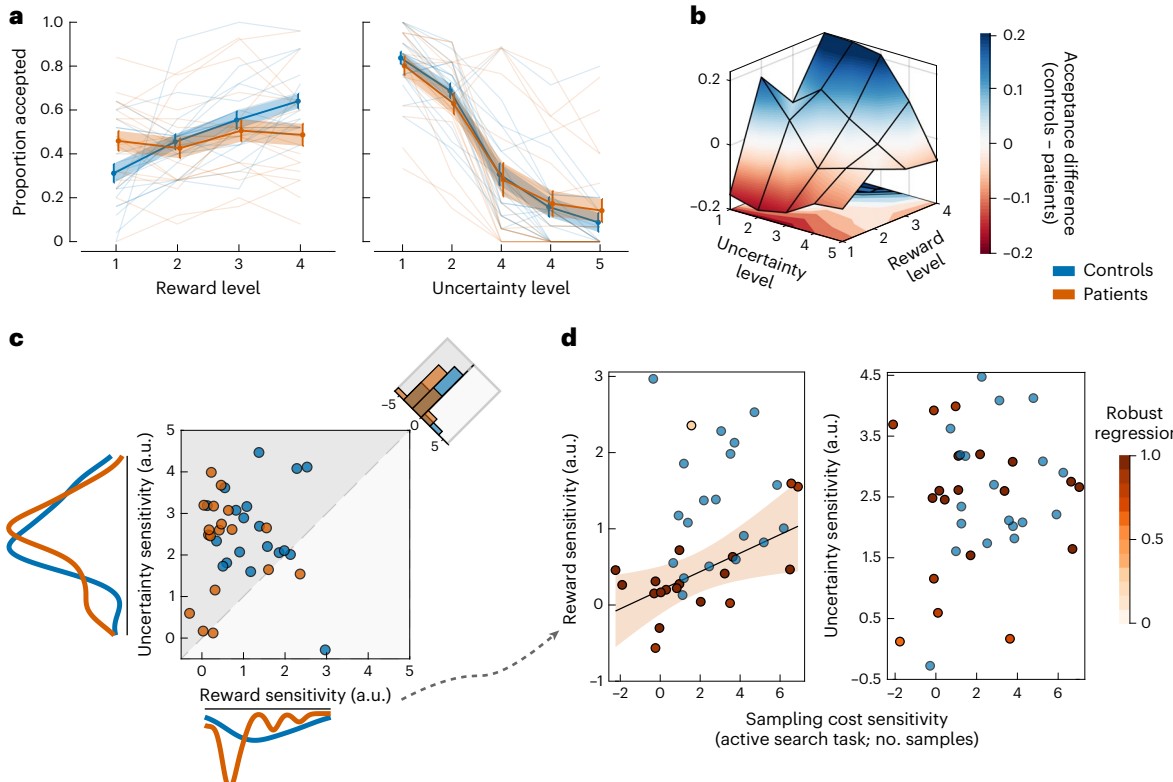

**Fig. 4 | Exp. 2—ALE patients are less sensitive to changes in reward under uncertainty. a**, Patients ($n = 19$) and healthy controls ($n = 19$) adjusted their decisions according to the reward and uncertainty on offer. The influence of reward on offer acceptance was blunted in ALE patients when compared with controls (shallower reward slope). Level 1 indicates lowest reward/uncertainty level on offer. **b**, Controls accepted more of the high-value offers (blue region) than hippocampal patients did. **c**, Investigation of the group effect using an $L_gMM$ revealed that patients had significantly lower sensitivity to reward than controls but did not significantly differ in their sensitivity to uncertainty (ALE × reward: $\beta = -0.983$; $t_{3748} = -3.58$; $P < 0.001$; 95% CI, (−1.52, −0.44)). It also showed that the

impact of uncertainty on decision-making was more significant than the impact of reward (histogram on the corner). **d**, Lower sensitivity to reward, but not to uncertainty, in the passive task is associated with lower sensitivity to sampling cost in the active sampling task (Exp. 1) driving group differences in the number of samples collected. The colour scale indicates the contribution of each data point to the model. The blue dots represent controls and are added for visual comparison. The error bars and shading in **a** show ±s.e.m. The solid line and shading in **d** show the regression line and 95% CI. See Supplementary Table 8 for the full statistical details.

they tended to under- or oversample (Fig. 3b). Optimal sampling refers to the number of samples, $s^*$, that maximizes the expected return, given the current cost–benefit structure ($R_0$, $\eta_s$, $\eta_e$) and search efficiency (that is, the rate at which participants reduce uncertainty from one sample to the next, parameterized as the information extraction rate, $\alpha$; Methods).

Both ALE patients and healthy controls oversampled when the sampling cost was high (deviation from optimal; ALE: $\beta = 3.93$; $t_{1138} = 4.32$; $P < 0.0001$; 95% CI, (2.15, 5.72); controls: $\beta = 1.93$; $t_{1138} = 3.485$; $P < 0.001$; 95% CI, (0.84, 3.02); see also Supplementary Table 5), but patients oversampled to a greater extent than controls when the initial reward reserve was high in these conditions ($z = 2.267$, $P = 0.023$, Cliff's $\delta = 0.43$; Fig. 3b). There was no significant difference between the two groups when the sampling cost was low ($t_{36} = 0.579$, $P = 0.56$).

ALE patients' sampling speed was also less deterred by increasing sampling cost than that of controls (ALE × $\eta_s$: $\beta = -0.0719$; $t_{2272} = -2.14$; $P = 0.033$; 95% CI, (−0.13, −0.005); Fig. 3c and Supplementary Table 4). This suggests faster and less deliberate sampling in ALE patients when sampling costs are high, as seen in a speed–efficiency trade-off characterizing sampling behaviour in both groups (inter-sampling interval ∝ $\alpha$; ALE: $\beta = 0.26$; $t_{1138} = 4.82$; $P < 0.0001$; 95% CI, (0.15, 0.36); controls: $\beta = 0.137$; $t_{1138} = 2.96$; $P = 0.003$; 95% CI, (0.04, 0.22); Fig. 3d and Supplementary Table 6).

These findings suggest that ALE patients' sampling behaviour demonstrates at least partially blunted sensitivity to sampling cost

leading to oversampling (that is, giving up more reward than needed in exchange for information) as well as faster sampling (see Supplementary Information for a computational model characterizing this difference in sensitivity to sampling cost; Supplementary Fig. 1). This had consequences in terms of total reward received, as patients' scores suffered to a greater extent than controls' when the sampling cost increased (ALE × $\eta_s$: $\beta = -3.66$; $t_{2272} = -2.00$; $P = 0.046$; 95% CI, (−7.26, −0.06); Supplementary Table 4).

### Exp. 2—ALE patients are less sensitive to changes in reward against uncertainty

The passive task paradigm (Exp. 2) offers a reliable mechanistic delineation between responses to reward and to uncertainty when making value-based decisions. This helps answer whether oversampling in ALE patients is indeed related to lower sensitivity to changes in reward, rather than increased sensitivity to uncertainty.

A generalized logistic mixed-effects model ($L_gMM$) with maximal randomness was used to analyse the accept/reject choice data (Supplementary Table 8). As expected, participants (patients and controls) adjusted their decisions rationally according to offer attributes, accepting more offers with higher rewards and lower uncertainty (main effect of reward on offer acceptance: $\beta = 1.41$; $t_{3748} = 7.15$; $P < 0.001$; 95% CI, (1.02, 1.79); main effect of uncertainty on offer acceptance: $\beta = -2.73$; $t_{3748} = -8.72$; $P < 0.001$; 95% CI, (−3.34, −2.11); Fig. 4a,b). There was no significant interaction between the effects of reward and uncertainty

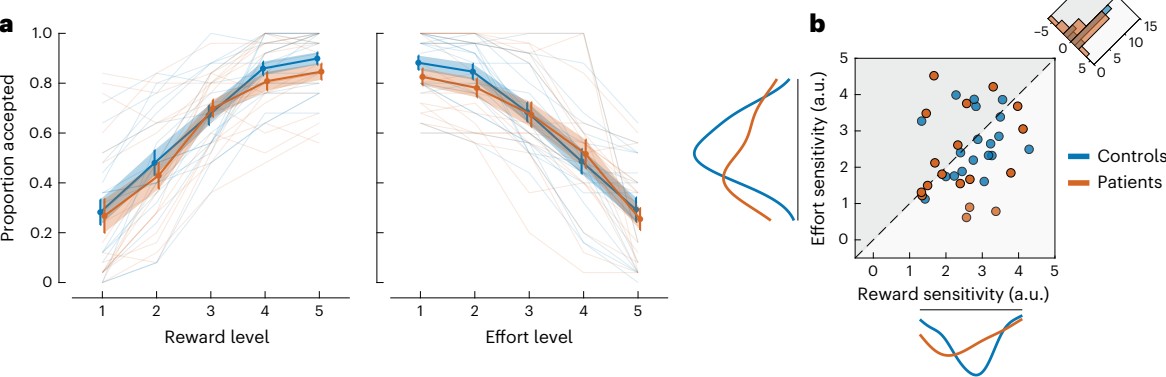

**Fig. 5 | Exp. 3—intact reward valuation against effort in ALE patients. a**, There was no significant difference in offer acceptance between patients ($n = 19$) and controls ($n = 19$). **b**, The estimates of these acceptance slopes indicate how sensitive participants are to reward/effort changes of offers. These sensitivity estimates were extracted from a generalized mixed model with full randomness and plotted for visualization. The error bars show ±s.e.m. See Supplementary Tables 10 and 11 for the full statistical details.

($R × EE: \beta = 0.065$; $t_{3748} = 0.53$; $P = 0.60$; 95% CI, (−0.17, 0.31)). Patients and controls did not significantly differ in the time they took to accept ($z = −0.379$, $P = 0.70$) or reject offers ($t_{36} = −0.34$, $P = 0.73$). However, they differed with regard to the influence that reward exerted on the decision to accept the offer. Compared with controls, ALE patients were overall less influenced by the reward on offer (ALE × reward: $\beta = −0.983$; $t_{3748} = −3.58$; $P < 0.001$; 95% CI, (−1.52, −0.44); Fig. 4a–c). By contrast, their sensitivity to uncertainty was not significantly different from controls' (ALE × EE: $\beta = 0.336$; $t_{3748} = 0.76$; $P = 0.45$; 95% CI, (−0.53, 1.20)). Taken together, this means that patients accepted fewer of the high-value offers (high-reward and low-uncertainty) ($z = −2.14$, $P = 0.032$, Cliff's $\delta = 0.55$ and $= −2.93$, $P = 0.003$, Cliff's $\delta = 0.40$, for the two offers of the highest value; Fig. 4b).

Next, in an exploratory analysis, we investigated whether sensitivity to changes in reward and uncertainty was associated with differences observed in sampling behaviour in Exp. 1. Sensitivity to reward and uncertainty were extracted from an $L_g$MM that included the two attributes and their interaction as predictors of offer acceptance (that is, the same model used above but with no group effect). For each participant, reward sensitivity and uncertainty sensitivity correspond to model-derived parameter estimates that capture how decisions are influenced by changes in these two offer attributes. A robust regression model showed that reward sensitivity, but not uncertainty sensitivity, correlated significantly with the sensitivity to sampling cost at high initial reward reserves in the active task—that is, the effect differentiating patients from controls in Exp. 1 (indexed by the difference in the number of samples between low- and high-sampling-cost conditions) ($R^2 = 0.33$; $\beta = 2.61$; $t_{17} = 2.88$; 95% CI, (0.70, 4.53); $= 0.010$; Fig. 4d). However, as this is an exploratory and potentially underpowered analysis, this correlation should be interpreted cautiously and might not be replicable.

In brief, the findings from Exps. 1 and 2 converge to indicate that ALE patients are less responsive to changes in reward under conditions of uncertainty in active and passive contexts. This is evidenced by less flexible sampling in response to changes in sampling cost in the active task, and reduced sensitivity to changes in the reward on offer in the passive task.

### Exp. 3—intact effort-based decision-making in ALE patients

In Exp. 3, we asked whether the blunted reward sensitivity observed in ALE patients for decisions involving a trade-off with uncertainty was also evident for a different discounting attribute—physical effort. In other words, is this a generalized phenomenon? A new version of a well-validated paradigm measuring effort and reward sensitivity was used to examine effort-based decision-making (Methods).

An $L_g$MM with full randomness was used to analyse the choice data (Supplementary Table 10). The results showed that, as expected, participants from both groups accepted more offers (showed more willingness to allocate effort) when the reward on offer increased and the required effort decreased (main effect of reward on offer acceptance: $\beta = 2.97$; $t_{4739} = 11.12$; 95% CI, (2.44, 3.49); $P < 0.0001$; main effect of physical effort on offer acceptance: $\beta = −2.82$; $t_{4739} = −8.45$; 95% CI, (−3.47, −2.16); $P < 0.0001$; Fig. 5a and Supplementary Table 10). However, neither reward nor effort sensitivity differed significantly between patients and controls (ALE × reward and ALE × effort, both $P > 0.05$; Fig. 5a,b). To quantify the evidence in favour of this null result, we ran the same analysis using Bayesian mixed modelling. Once again, this analysis did not suggest differences between the two groups in reward (or effort) valuation (Bayes factor of 3.78 for null effect of ALE × reward; Bayes factor of 5.28 for null effect ALE × effort; Supplementary Fig. 4 and Supplementary Table 11). Note that there was also no significant difference in total acceptance of offers (effect of disease (ALE) on offer acceptance: $\beta = −0.13$; $t_{4739} = −0.13$; 95% CI, (−1.356, 1.082); $P = 0.82$) or in decision times (accept decisions: $t_{36} = 0.87$; 95% CI, (−0.22, 0.56); $P = 0.388$; Cohen's $d = 0.27$; reject decisions: $t_{36} = 0.99$; 95% CI, (−0.16, 49); $P = 0.325$; Cohen's $d = 0.31$). Compared with Exp. 2, these decision times were slower (Exp. 2: $\mu = 2.04$, s.d. = 0.40; Exp 3: $\mu = 2.40$, s.d. = 0.49; $t_{18} = 2.25$; 95% CI, (0.02, 0.68); $P = 0.036$; Cohen's $d = 0.77$), indicating less deliberation when making decisions under uncertainty in Exp. 2 than when making effort-based decisions (Supplementary Fig. 2 and Supplementary Table 24; see Supplementary Information for an extended analysis of decision times across Exps. 2–4).

Finally, the group effect on reward sensitivity across Exps. 2 and 3 was examined using a generalized mixed model after combining choices from both tasks (Supplementary Tables 12 and 13). As expected, reward sensitivity in ALE patients compared with controls was significantly blunted in Exp. 2 compared with Exp. 3 (ALE × reward × task: $\beta = 0.467$; $t_{8495} = 4.15$; 95% CI, (0.24, 0.68); $P < 0.0001$; Supplementary Table 12). These results are consistent with a sparing of reward sensitivity in ALE patients when reward had to be weighed against effort, without uncertainty being considered.

### Exp. 4—blunted reward and effort sensitivity under uncertainty in ALE patients

One might argue that the comparison of reward sensitivity in Exps. 2 and 3 is not fully matched due to the difference in task cues and environment (for example, reward as credits versus virtual apples). These concerns were addressed using a new version of Circle Quest that was designed to examine effort-based decision-making under

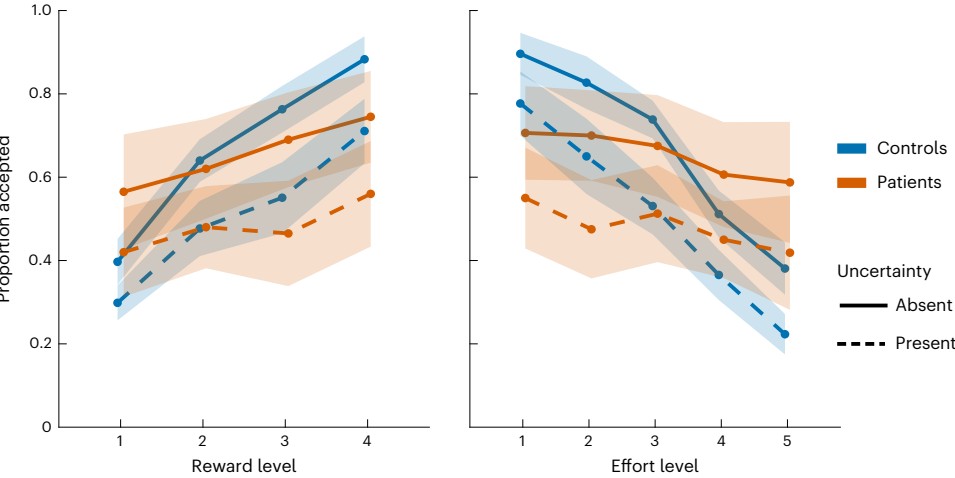

**Fig. 6 | Exp. 4—blunted reward and effort sensitivity under uncertainty in ALE patients.** In Exp. 4, the participants were required to accept or reject offers taking into consideration three attributes: reward, physical effort and uncertainty. The results showed that ALE patients ($n = 8$), compared with controls ($n = 12$), were less sensitive to changes in reward under uncertainty and effort (the slope difference between the continuous and dashed lines across the two groups) while having intact uncertainty sensitivity (the degree of downward shift between the continuous and dashed lines) (reward × uncertainty × ALE: $\beta = 0.684$; $t_{4184} = 2.37$; 95% CI, (0.11, 1.24); $P = 0.018$; effort × uncertainty × ALE: $\beta = -0.541$; $t_{4184} = -1.97$; 95% CI, (−1.07, −0.003); $P = 0.049$). These results are consistent with the findings from Exps. 1 and 2, highlighting disrupted reward and cost valuation in ALE patients under uncertainty. The shaded areas show ±s.e.m. See Supplementary Table 14 for the full statistical details.

uncertainty (Fig. 2). The task had the same reward levels and cues used in Exp. 2 and the same effort levels and cues used in Exp. 3. The participants (12 controls and 8 ALE patients) also made the decisions either with or without uncertainty. Thus, the task overall examined how participants adjusted their decisions when taking into consideration these three attributes (reward, effort and uncertainty).

Choice data were analysed using the same approach as in Exps. 2 and 3 with generalized mixed-effects models (Supplementary Table 14). The results showed that ALE patients were indeed less sensitive to reward than controls when making effort-based decisions under uncertain conditions (reward × uncertainty × ALE: $\beta = 0.684$; $t_{4184} = 2.37$; 95% CI, (0.11, 1.24); $P = 0.018$).

The three-way interaction investigating group difference in effort sensitivity under uncertainty was also statistically significant (effort × uncertainty × ALE: $\beta = -0.541$; $t_{4184} = -1.97$; 95% CI, (−1.07, −0.003); $P = 0.049$; Fig. 6 and Supplementary Table 14). There was no significant difference between the two groups in their sensitivity to uncertainty (ALE × uncertainty: $\beta = 0.89$; $t_{4184} = 1.003$; 95% CI, (−0.85, 2.65); $P = 0.315$).

These findings are thus consistent with the results from Exps. 1 and 2 showing intact sensitivity to uncertainty and reduced sensitivity to reward in the presence of uncertainty in ALE patients. They also suggest that effort sensitivity can be blunted in ALE patients under conditions requiring uncertainty consideration. Thus, overall, while healthy controls managed to flexibly adjust their decisions taking into consideration the three attributes of the offers, ALE patients were generally more responsive to the uncertainty component of the offers, with less emphasis on other economic attributes such as reward and effort. Importantly, this is not a replication of the Exp. 2 results because (1) decision-making involves effort consideration in addition to uncertainty and reward and (2) the task includes a new certain condition against which the effect of uncertainty is compared. That said, these findings from Exp. 4 should interpreted with caution due to the small sample size.

In summary, the results from Exps. 1–4 suggest that the presence of uncertainty is associated with deficits in processing other attributes (reward and effort) in hippocampal dysfunction, with intact uncertainty processing.

## Hippocampal atrophy correlates with decreased reward sensitivity under uncertainty

A whole-brain voxel-based morphometry (VBM) analysis was performed. Compared with healthy controls, ALE patients had lower grey matter intensity in three clusters involving the limbic, thalamic and temporal regions (Supplementary Table 15). As expected, the largest cluster (limbic region) included mainly the hippocampal and parahippocampal regions (Fig. 7a). Hippocampal atrophy was also demonstrated by comparing extracted total hippocampal volume using the Freesurfer analysis pipeline, showing reduced right whole hippocampal volumes in ALE patients (Supplementary Table 1). Note that a few participants also had severely atrophied left hippocampi (with and without right hippocampal atrophy; Supplementary Table 2).

We then performed robust regression analysis to examine the relationship between hippocampal atrophy and reduced reward sensitivity observed in ALE patients. For this purpose, reward and uncertainty sensitivities from Exp. 2 were used as key behavioural markers characterizing the performance of ALE patients when compared to controls. The results showed that sensitivity to changes in reward was associated with total average hippocampal volumes in patients (model $R^2 = 0.41$; $\beta = 0.0006$; $t_{13} = 3.05$; 95% CI, (0.0002, 0.001); $P < 0.009$; Fig. 7b). This correlation remained significant after we controlled for age and gender (model $R^2 = 0.38$; $\beta = 0.0006$; $t_{11} = 2.28$; 95% CI, (0, 0.0012); $P = 0.043$). In contrast, the correlation between uncertainty sensitivity and total hippocampal volume was not significant (model $R^2 = 0.062$; $\beta = 0.0004$; $t_{13} = 0.92$; 95% CI, (−0.0006, 0.001); $P = 0.371$).

Repeating the same analysis for reward sensitivity from Exp. 3 revealed no significant correlation between total hippocampal volume and reward sensitivity against effort ($P = 0.88$). Due to the limitation of sample size, this analysis was not performed for Exp. 4.

To assess the anatomical specificity of this result, we also extracted the volume of the amygdala—another limbic brain region that was highlighted in the VBM analysis—and the same analysis was run on this metric. This showed no significant correlation between the amygdala volume and sensitivity to reward or uncertainty (Supplementary Fig. 5).

## Intact localization and uncertainty estimation in ALE patients

Two control analyses were performed to examine possible factors that might influence response to uncertainty in the Circle Quest task. First,

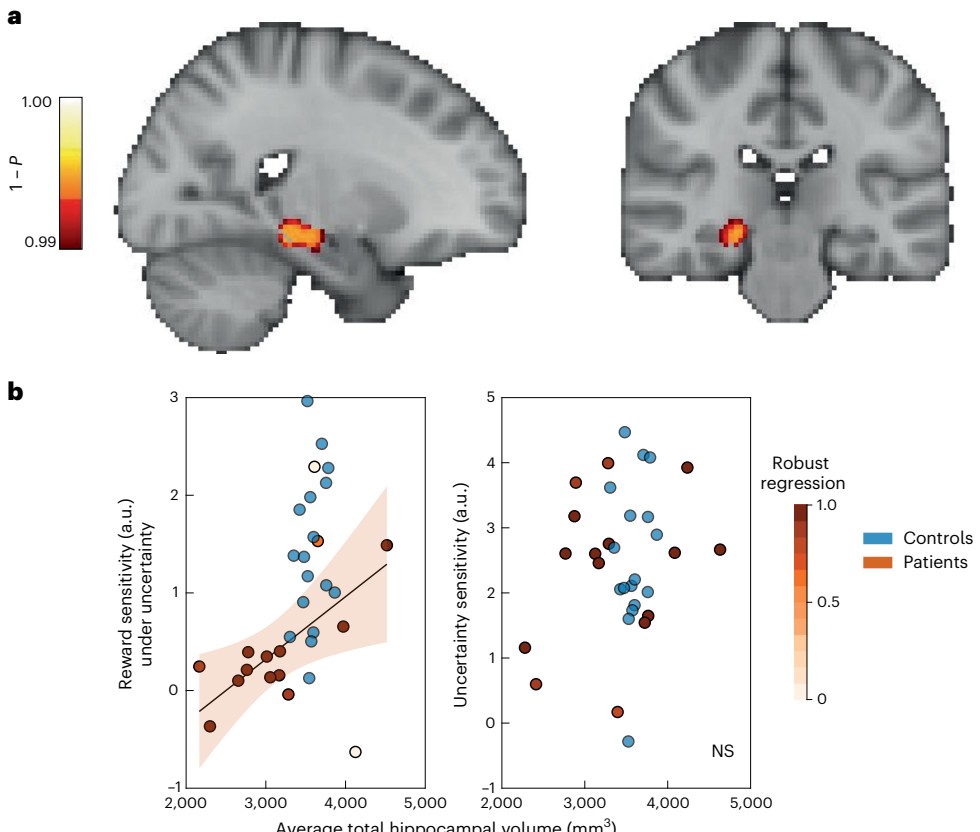

**Fig. 7 | Severity of hippocampal atrophy correlates with decreased reward sensitivity under uncertainty. a**, VBM analysis shows that ALE patients have significantly reduced grey matter intensity in the right hippocampal region (cluster 1, Supplementary Table 15). **b**, Patients with more severely atrophied hippocampi were less sensitive to reward when traded against uncertainty (behavioural data from Exp. 2). By contrast, hippocampal volumes were not significantly correlated with sensitivity to uncertainty, which was preserved in ALE patients (model $R^2 = 0.41$; $\beta = 0.0006$; $t_{13} = 3.05$; 95% CI, (0.0002, 0.001); $P < 0.009$). The shading around the regression line represents the 95% CI. NS, not significant.

participants might differ in their motor performance and accuracy of placing the blue disk for given uncertainty levels, which might bias their estimation and valuation. To examine this, we analysed performance in the training task where the participants were required to move the blue disk for fixed levels of uncertainty. Distance to the hidden circle was compared between the two groups. There was no significant difference between ALE patients and controls in this metric, indicating similar placement performance on the task (difference between ALE patients and controls in Exps. 1 and 2: $z = 1.10$, $P = 0.267$, Cliff's $\delta = 0.21$; Exp. 4: $z = 0.10$, $P = 0.913$, Cliff's $\delta = 0.038$; Fig. 8a). To compare localization performance on trials that did not feature the same levels of uncertainty, we calculated the distance to the optimal placement point (the centroid of the posterior belief of all the possible locations of the hidden circle). This measure across all versions of Circle Quest was not significantly different between the two groups (Supplementary Fig. 6).

Second, different participants might have different estimations of uncertainty for the same visual displays, ultimately affecting their performance on the task as they gather information and make decisions. To examine inter-individual differences in uncertainty estimation, the participants provided these estimates as confidence ratings before making decisions in Exp. 2 and during training in Exp. 4 (Methods). A generalized mixed-effects model indicated that in both experiments there was no significant difference in subjective uncertainty estimation between patients and controls (Exp. 2, ALE × EE: $\beta = -0.054$; $t_{3752} = -0.77$; 95% CI, (−0.19, 0.08); $P = 0.442$; Exp. 4, ALE × EE: $\beta = -0.022$; $t_{2306} = -0.16$; 95% CI, (−0.30, 0.25); $P = 0.873$; Fig. 8b and Supplementary Table 16). The results indicate that subjective estimates of uncertainty mapped well onto experimentally defined uncertainty across study participants.

As a result, the choice performance results did not change when these subjective estimates were used to analyse performance instead of objective uncertainty (EE) (Supplementary Fig. 7 and Supplementary Table 18).

In addition to these control analyses, we investigated the possible effects of cognitive deficit and memory decay on performance. There was no significant correlation between cognitive performance indexed by ACE III scores (whether total or subdomains) and sensitivity to either reward or uncertainty in Exp. 2 (robust regression: ACE III ∝ reward sensitivity in Exp. 2: $R^2 = 0.18$; $\beta = 0.01$; $t_{17} = 0.59$; 95% CI, (−0.02, 0.05); $P = 0.557$; ACE III ∝ uncertainty sensitivity: $R^2 = 0.028$; $\beta = 0.03$; $t_{17} = 0.68$; 95% CI, (−0.075, 0.14); $P = 0.501$; Supplementary Fig. 8a and Supplementary Table 20).

To investigate whether performance (mainly reward sensitivity) was influenced by task duration across the different tasks in ALE patients, we analysed decisions in the first half compared to the second half in Exps. 2–4. This showed that reward sensitivity across the different tasks did not differ between the two task halves in ALE patients, indicating minimal effect of memory decay during task performance (reward × second task half, Exp. 2: $\beta = 0.21$; $t_{1848} = 1.16$; 95% CI, (−0.14, 0.56); $P = 0.246$; Exp. 3: $\beta = 0.35$; $t_{2364} = 1.39$; 95% CI, (−0.14, 0.85); $P = 0.163$; Exp. 3: $\beta = -0.039$; $t_{1592} = -0.25$; 95% CI, (−0.35, 0.27); $P = 0.803$; Supplementary Fig. 8b and Supplementary Table 22).

Finally, we analysed the catch trials from Exp. 4 to investigate features that might suggest random responding (for example, sensitivity to sudden changes in uncertainty levels). The results from these trials showed that ALE patients responded as expected with intact sensitivity to sudden uncertainty changes during both the uncertainty

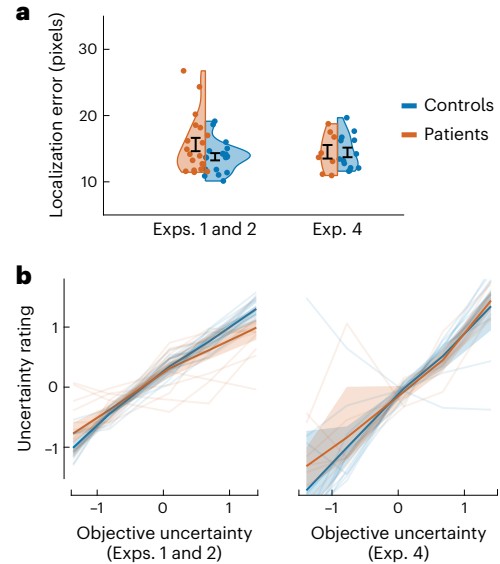

**Fig. 8 | Exps. 1, 2 and 4 — intact localization and subjective uncertainty estimation in ALE patients. a**, ALE patients ($n = 19$ in Exps. 1 and 2, $n = 8$ in Exp. 4) and controls ($n = 19$ in Exps. 1 and 2, $n = 12$ in Exp. 4) did not differ in their localization error (the distance between the centre of the blue disk and the hidden circle) for fixed levels of uncertainty, indicating similar motor and localization performance. **b**, Subjective uncertainty is measured as $z$-scored sign-flipped confidence ratings that participants reported before seeing the reward on offer (Exp. 2) or during training (Exp. 4, including catch trials). There was no significant difference between ALE patients and controls in this measure, indicating intact uncertainty estimation. The error bars in **a** show ±s.e.m. The solid lines and shading in **b** show mean values and 95% CIs. See Supplementary Table 16 for the full statistical details.

estimation phase (ALE × uncertainty: $\beta = 0.0056$; $t_{206} = 0.05$; 95% CI, (−0.22, 0.23); $P = 0.960$) and the decision phase (ALE × uncertainty: $\beta = -0.914$; $t_{201} = -0.20$; 95% CI, (−9.84, 8.01); $P = 0.840$; Supplementary Fig. 8c and Supplementary Table 23). ALE patients also showed blunted sensitivity to effort during these trials, as demonstrated in the rest of the experiment (ALE × effort: $\beta = 6.63$; $t_{201} = 2.74$; 95% CI, (1.85, 11.41); $P = 0.006$). Analysis of reward responsivity in these trials is restricted because of their unbalanced design, which does not feature all reward levels equally, as well as the small sample size.

## Discussion

The studies presented here assessed patients with ALE, which is associated with damage to the hippocampus. In four experiments, patients were assessed on how they evaluated reward, uncertainty and effort when making value-based decisions. The results converged to indicate that ALE patients had intact uncertainty processing across different contexts such as passive reward valuation (Figs. 4 and 6) and active information gathering prior to decisions (Fig. 3). However, whenever uncertainty was a factor that needed to be considered, sensitivity to other attributes (reward and physical effort) was blunted (Figs. 4 and 6), despite intact valuation of these attributes in a context that did not feature uncertainty (Fig. 5). Reduced sensitivity to changes in reward in the context of uncertainty correlated with the severity of hippocampal atrophy in ALE patients (Fig. 7). The results thus indicate a specific role of the hippocampus in processing uncertain rewards and costs.

Previous decision-making studies have demonstrated several potential contributions of the hippocampus to goal-directed behaviour. For example, numerous reports have indicated that the hippocampus might be critical for inter-temporal decision-making, especially when participants are required to imagine future outcomes[47–49]. The results of other investigations have demonstrated that the

hippocampus might be needed for deliberation preceding value-based decisions but not other value-independent decisions such simple sensory discrimination[50]. This differential involvement proposes a specific role of the hippocampus in value construction and estimation during decisions rather than general executive processing. In line with this, some researchers have demonstrated hippocampal involvement in inferring values of novel stimuli from previously encountered cues and stimuli[4,5,7]. These functions are thought to be related to the established role of the hippocampus in episodic thinking and prospection, with some researchers proposing that it contributes to inferring values through sampling from memories or futuristic projections[51–55].

In a broader conceptual context, such scenarios necessitating value inference based on mental time travel can be regarded as forms of decision-making under uncertainty. For instance, examining delay discounting in inter-temporal decisions reveals cognitive and computational parallels with probabilistic discounting that characterizes uncertainty valuation[56–60]. This resemblance might stem from the inherent risk associated with both discounting properties, representing the likelihood of obtaining probabilistic or delayed rewards[57,59]. But it might also reflect a temporal aspect in probabilistic discounting, where agents consider the attempts required to secure certain probabilistic rewards and the time investment[58]. The hippocampus seems to have a pivotal role in inter-temporal decision processes that particularly involve episodic future thinking, where the inclusion of episodic details of future rewards reduces delay discounting[16,61,62]. Hippocampal damage in humans has been shown to reduce this effect and blunt sensitivity to delay changes and future rewards requiring episodic inputs[16,63]. Such effects could also be interpreted from an uncertainty-centred view, where the hippocampus's episodic future thinking contributes to reducing the uncertainty attached to these values or delays.

This conceptualization is consistent with functional neuroimaging studies that have demonstrated hippocampal activation that correlates with the degree of uncertainty (entropy) of sensory stimuli when making decisions[23–26,64] as well as measures of reactivity to uncertainty (for example, information sampling speed) prior to committing to decisions under uncertainty[22]. These investigations align with the view that regards uncertainty as a threatening stimulus (that is, one that carries risk signals) processed by the hippocampus-centred behavioural inhibition system[65]. The hippocampus, according to this view, works as a mismatch-detection system comparing expectations with perceived stimuli and triggering behavioural avoidance when confronted by uncertainty (or other anxiety-inducing stimuli)[65].

These features of uncertainty as a probabilistically discounted risk signal offer a key distinction from other costs such as physical effort, which is more deterministic. The results presented in this study provide evidence that the hippocampus is critically involved in valuation processes under the first context (uncertainty) but not the second (physical effort), providing specific insights into the role of the hippocampus in value-based decision-making. Counterintuitively, hippocampal damage was associated with the preservation of uncertainty estimation and valuation, but blunted sensitivity to other decision attributes (for example, reward and effort) if simultaneously considered with uncertainty.

These findings suggest that hippocampal damage might be related to a specific deficit in the integration of relatively intact uncertainty signals with other attributes that contribute to the value space. Note that this is unlikely to reflect a global deficit in decision-making or value computation, as participants demonstrated intact effort-based decision-making in Exp. 3 (Fig. 5). One possible explanation for this might be that people with hippocampal atrophy possess limited computational resources that prioritize uncertainty processing (and perhaps other risk-related signals) over other value determinants when computing subjective values to guide behaviour and decisions. In fact, the results from the active search experiment (Exp. 1) could also be interpreted as evidence of uncertainty prioritization, as ALE

patients sampled faster and more extensively than controls to abolish uncertainty before committing to their decisions (Fig. 3c), disregarding changes in sampling cost. The hippocampal role therefore might be to infer decision values by multiplying the probability of outcomes (that is, uncertainty) with other economic attributes such as reward and effort. These signals might then be related to other regions involved in subjective value estimation such as the anterior cingulate cortex and orbitofrontal cortex[5,50,66–73].

These findings resonate with previous reports that highlighted the hippocampus as part of a wider brain valuation network, which includes regions that have established roles in processing rewards and costs such as dopaminergic brain regions (for example, the ventral striatum), the ventromedial prefrontal cortex and the anterior cingulate cortex[74]. The hippocampus shares functional and structural connections with these regions and is thought to provide contextual information that supports the valuation process[5,50,66–70]. The behavioural results in this study indicate that hippocampal processing of uncertainty signals might be a key determinant of how different brain regions evaluate rewards and costs when computing subjective values.

The neural mechanism underlying how hippocampal signals support reward-related behaviours is not yet fully established and will require further research. However, previous studies have demonstrated that the hippocampus might share future-related signals (for example, preplay and look-ahead signals) with dopaminergic regions involved in reward evaluation and processing[13–15,75]. Similarly, several forms of reward representation in the hippocampal formation and its extended networks have been reported, correlating with various reward-related behaviours and cognitive processes such as appetitive responses to rewards[76,77], approach–avoid decisions[78] and reward-guided exploration[1]. While hippocampal-lesioned animals might show behavioural adjustments in responses to changes in reward in the environment[79,80], these findings might differ between contexts and depend on the way reward is manipulated (for example, whether animals move from high-reward environments to low, or the opposite)[77]. Consummatory responses of lesioned animals seem to be unaffected despite the changes in behaviour, challenging the notion that the role of the hippocampus in reward processing is of a hedonic nature[80], and further substantiating its goal-related and context-dependant contribution.

This uncertainty-sensitive hippocampal contribution might serve to support not only passive valuation but also active goal-directed behaviours such as information seeking to resolve uncertainty. While results from a previous investigation have shown that people with hippocampal damage exhibit less structured and stochastic visual information exploration than controls[27], it was unclear how economic factors come into play. In this study, less optimal decision-making when uncertainty-related value computations had to occur was evident in ALE patients when they actively gathered information and sequentially updated the expected values of their decisions, as well as when they made passive decisions under uncertainty. The source of this disruption was related to less flexible decision adjustments based on reward changes in the environment while maintaining sensitivity to uncertainty. The results thus provide insights into how the hippocampus contributes to the economics of information gathering, in addition to its potential exploratory role[81], which was not strongly affected in ALE patients as they managed to reduce uncertainty as efficiently as healthy controls.

A positive correlation was observed between total hippocampal volumes and reward sensitivity under uncertainty. Future investigations might build on this to investigate the drivers of this relationship. For example, such an association might be related to specific hippocampal subfields and sub-circuits. In rats, reward cells have been described in the subiculum and CA1 subfields[82], which might suggest that atrophy or disruption of these regions might have a more specific role in the process. Moreover, distinct hippocampal subfields and regions might have differential functional connectivity profiles with other cortical and sub-cortical regions that might be contributing to

motivation and decision-making[22,83–86]. Hippocampal damage observed in ALE patients is likely to be associated with more widespread disturbances of such networks contributing to reward valuation under uncertainty[35,87–89]. With advanced neuroimaging acquisition and analysis techniques, it will be crucial to try to answer these questions at the subfield level and provide a more comprehensive account of hippocampal contribution to motivation.

It is, however, challenging to fully ascertain whether the results of this study reflect a specific computational property of the hippocampus or instead a general disruption of cognitive processing that might be observed with other brain lesions. Three factors make this possibility unlikely. The first is the correlation between behaviour and the severity of hippocampal atrophy, rather than other closely related regions such as the amygdala, which might have been affected by the disease process as well (Fig. 7 and Supplementary Fig. 5). Second, the results do not correlate with cognitive dysfunction indexed by ACE III scores (both total and subdomains, including memory) or metacognitive deficits in uncertainty estimation (Supplementary Fig. 8a and Supplementary Tables 19 and 20). Third, the analysis of additional experimental parameters, including performance on catch trials and closer examination of decisions made around task onset and finish points (Supplementary Fig. 8b,c), contradicts the idea that performance is a reflection of cognitive dysfunction or random responding. This is especially notable considering that the main results exhibit a selective deficit not globally affecting all value attributes, as one would expect with chance-level performance. Moreover, the task design and administration have been tailored to minimize such effects, including completing comprehension and debriefing questionnaires, using interactive tutorials for training and adding cues to reduce memory load. Nevertheless, despite these considerations, it might be impossible to completely rule out the effect of cognitive or memory deficits on decision-making in patients with hippocampal damage, and this could be a potential limitation of this study. It would be more reasonable to aim to interpret the findings within the broader context of hippocampal episodic and memory functions rather than in isolation, as highlighted in our previous discussions (for example, considering similarities between inter-temporal decisions and uncertainty).

In a similar vein, it could be argued that the results might merely reflect a difficulty effect imposed by more complex demands of uncertainty cues than those of other cues. However, if this were the case, one would expect patients to take longer than controls when making decisions under uncertainty (Exp. 2). This expectation did not align with the observed results. It would also be predicted to find slower reaction times in Exp. 2 (involving reward and uncertainty) than in Exp. 3 (involving reward and effort). On the contrary, the results demonstrated the opposite trend, indicating shorter deliberation time in ALE patients when confronted with uncertainty. This performance is consistent with the decision-making pattern observed in the study that tends to disregard other attributes in the presence of uncertainty, potentially leading to quicker decisions centred on uncertainty. A similar tendency is also evident in the active sampling experiment (Exp. 2), with faster sampling rates among ALE patients than among controls. While such analyses indirectly provide some insights into complexity effects on performance, future studies might want to focus on disentangling this experimentally in task designs that feature analogous cues for the attributes being measured. Along these lines, it would also be insightful to contextualize and establish the empirical connection between the observed pattern of the results in this study and a broader spectrum of potential hippocampal roles in goal-directed behaviour, such as task representation from integrating complex features[70,90,91] and experimental range adaptation[92]. Impairment in these functions might lead to diminished sensitivity to some task features when agents are required to integrate them to guide decisions and behaviour[63,93,94].

The present study has a few other limitations. First, it is possible that the dissociation between effort and uncertainty in Exp. 2 and

Exp. 3 might be not well delineated. For example, in the Circle Quest paradigm, one might need to consider effort costs such as moving the blue localization disk on the screen or additional cognitive effort required to translate the configuration of dots into uncertainty estimates. Such costs can also affect active sampling behaviour, influencing the speed and efficiency of gathering information[39] (see also Supplementary Information for a computational model characterizing these effects). Similarly, the effort task (Exp. 3) might theoretically involve some elements of uncertainty (for example, aligning visualized effort levels to subjective cost estimates). Second, while Exp. 4 was designed to address such limitations, only a subset of the participants performed it, which might not be strictly representative of the larger group. This task also introduced a more complex decision structure that might require more planning (two steps to reach goals) and additional cognitive load on memory and attention. It is important to discuss and consider the results from one experiment in the context of the other experiments performed to obtain a bigger picture of how the hippocampus might be contributing to value processing. Finally, it might also be beneficial for future studies to obtain more objective measures of such processes (for example, pupillometry for reward sensitivity[40,95]), which could help bridge the gap with conclusions based on subjective estimates obtained from decision-making and goal-directed behaviour.

In conclusion, the results presented here provide evidence from human participants that the hippocampus plays a crucial role in decision-making. This contribution appears to be specific to contexts that involve uncertainty, influencing how people evaluate other economic attributes such as reward and effort.

## Methods

### Participants

In Exps. 1–3, 19 individuals with a previously established diagnosis of ALE (age: $\mu = 60.00$ years, s.d. = 11.36 years, 13 males) were tested along with 19 healthy age- and gender-matched controls (age: $\mu = 61.16$ years, s.d. = 11.71 years, 13 males). In Exp. 4, 8 ALE patients and 12 controls completed an additional follow-up task. The sample size was determined on the basis of previous comparable work in ALE patients[34,36,37] as well as previous research using the behavioural paradigms used in the study[22,39,40,45]. All participants gave written consent to take part in the study and were offered monetary compensation for their time. The study was approved by the University of Oxford ethics committee (RAS ID No. 248379, Ethics Approval Reference No. 18/SC/0448). Supplementary Tables 1 and 2 show the demographics and characteristics of the study groups. Due to a technical error during testing, two blocks (out of five) were missing from the passive choices task for one patient (code 14 in Supplementary Table 2). Analyses were conducted with and without this patient's data, and there was no difference in the results or conclusions. After completing a practice session, the participants had to correctly answer all the questions of a task comprehension quiz to be eligible to do the task and continue the experiment.

### External measures

All participants underwent a cognitive assessment using ACE III[96] and an executive function assessment with digit span. They also completed self-report questionnaires on apathy (Apathy Motivation Index[97]), depression (Beck Depression Inventory-II[98]), fatigue (Fatigue Severity Scale[99]) and hedonic experience (Snaith–Hamilton Pleasure Scale[100]).

### Procedure

The tasks were presented on a 17-inch touchscreen PC using MATLAB (MathWorks, version 2018b) and Psychtoolbox[101,102] version 3. Testing was done in a quiet room with an experimenter present at all times. The participants sat within reaching distance of the screen (~50 cm) and were instructed to use the index finger of their dominant hand to respond. The task environment was adjusted according to handedness (for example, uncertainty rating on the side of the dominant hand in Exps. 2 and 3).

### Experimental paradigm for Exps. 1 and 2

A modified, shorter version of the Circle Quest task was used (described in detail in ref. 39). In Exp. 1, the participants performed the active sampling version of the task designed to test active information gathering prior to committing to decisions. In Exp. 2, the participants performed the passive choices/decisions version designed to test decision-making under fixed, experimentally defined levels of uncertainty and reward. The participants were told that their goal in the two parts of the task was to maximize reward. All participants performed a training task followed by the active task and then the passive task. The purpose of this order was to ensure that by the time the participants performed the passive choices version of the task, they had extensive training and exposure to the task environment and scoring function through their interaction with the task during the active version. The average total duration of the testing session was approximately one hour (average duration in minutes ± s.d.: training, 7.38 ± 2.96; active sampling, 30.11 ± 5.20; passive choices, 22.86 ± 1.48).

**Training and exposure.** The task was explained to the participants using an interactive tutorial in the presence of the experimenter. In this interactive tutorial, the participants were simply required to localize a hidden purple circle on the screen. This circle had a fixed size on all trials (radius, 130 pixels; area, 5.80% of the search space). Two purple circles of the same size were always present on the right and left sides of the screen as a visual reminder of the circle on quest. This served to limit the memory demands of the task. Clues about the location of the hidden circle could be obtained by touching the screen. If a purple dot appeared where they touched (radius, 4 pixels), this indicated that the location was inside the hidden circle. Alternatively, if a white dot appeared where they touched, the location was outside the hidden circle. After completing the search, they were asked to move a blue disk of the same size as the hidden circle on top of where they thought the hidden circle was located, on the basis of the information they had gathered. During this tutorial, the participants performed five trials with no constraints on the number of samples they could acquire and without any sampling penalty. They were also encouraged to ask the experimenter in case they had any questions.

Following this short introduction to the game, the participants performed a training task that included 20 trials. The goal of this task was to (1) practise localizing the hidden circle for various levels of uncertainty and (2) expose the participants to the scoring function. On each trial of this training task, a configuration of eight dots (four purple and four white) was displayed on the screen, and the participants were required to move the blue disk on top of where they thought the hidden circle was located. Different configurations mapped onto different levels of uncertainty. For example, displays with spaced-out purple dots represented lower levels of uncertainty than displays in which the purple dots were clustered more closely. The former configuration had a lower number of possible circle placements that were compatible with the dots displayed on the screen (purple dots should be inside the hidden circle), and consequently, the expected localization error (EE, a quantitative measure of uncertainty) for such a configuration was smaller. EE was calculated as the probability-weighted average of all possible errors a participant could incur by placing the blue disk at the best possible location (the centroid of posterior belief) (Fig. 1b; for more details, see Supplementary Information). The two circles on the right and left sides of the search space contained the reward at stake, which the participants could obtain if they managed to perfectly localize the hidden circle. After the participants placed the blue disk, the location of the hidden circle was revealed, and the score they obtained for this localization appeared. The score was calculated as the reward at stake minus the localization error penalty upon placing the blue disk ($\eta_e \times e$), where $e$ is the distance between the centre of the blue disk and the centre of the hidden circle in pixels, and $\eta_e$ is the spatial error cost per pixel, which was constant and equal to 1.2 credits per pixel.

**Active sampling task (Exp. 1).** In the active version of the task (Fig. 1), the participants could reduce their uncertainty about the location of the hidden circle by actively sampling the search space—that is, touching the screen to obtain information. Similar to the training tutorial, this provided them with binary information about the location of that sample in relation to the hidden circle. If where they touched was inside the hidden circle, the sample was purple. Otherwise, the sample was white. At the beginning of every trial, the participants had 18 seconds to sample whenever, wherever and how much they wanted. After this, they were required to move the blue disk to where they thought the hidden circle was located to collect monetary credits. The participants started each trial with an initial reward reserve, $R_0$, from which they lost credits each time they acquired a new sample depending on the cost of sampling, $\eta_s$, on that trial. There were two levels of initial reward ($R_0$: low, 95 credits; high, 130 credits) and two levels of sampling cost ($\eta_s$: low, 1 credit per sample; high, 5 credits per sample), giving rise to four conditions in four blocks that were counterbalanced between participants. Each condition included 15 trials. The score (in credits) that participants obtained on each trial was calculated as follows:

$$\text{Score} = R_0 - s \times \eta_s - e \times \eta_e \qquad (1)$$

where $R_0$ is the initial reward reserve, $s$ is the number of samples obtained, $\eta_s$ is the cost of acquiring a new sample, $e$ is the placement error and $\eta_e$ is the error cost per pixel, which was fixed and equal to 1.2 credit per pixel.

**Quantifying deviations from optimal behaviour.** The expected value, EV, changed dynamically with each sample, $s$, as follows:

$$\text{EV}(s) = R_0 - s \times \eta_s - \text{EE}(s) \times \eta_e \qquad (2)$$

where EV($s$) is the expected value at the $s$th sample, which is calculated as what is left of the initial reward reserve $R_0$ after subtracting the credits lost during sampling ($s \times \eta_s$) and the expected localization error penalty (EE($s$) $\times \eta_e$). The optimal number of samples to acquire corresponds to the number of samples, $s^*$, that maximizes the expected value. Deviation from optimal sampling (either over- or undersampling) is the difference between the number of samples obtained and $s^*$. Note that in the equation above, the rate at which EE decays from one sample to the next depends on the participant's choices of sampling locations (Fig. 3a). Therefore, the dynamics of EE over successive samples (that is, the efficiency of the search) may vary across trials and participants. Sampling efficiency was parameterized as the information extraction rate, $\alpha$, which was computed for each trial as follows:

$$\hat{\text{EE}}_{(n,1)} = \frac{\sum_{i=1}^{60} \text{EE}_{(i,1)}}{60}$$
$$\hat{\text{EE}}_{(n,s)} = (\hat{\text{EE}}_{(n,1)} - \hat{\text{EE}}_\infty) \times (1 - \alpha_n)^{s-1} + \hat{\text{EE}}_\infty \qquad (3)$$
$$0 < \alpha < 1 \text{ and } \hat{\text{EE}}_\infty > 0$$

where $n$ is the trial number, $s$ is the sample number within the trial (the 0th sample is the sample displayed on the screen at the beginning of the trial, before the participants touched the screen; Fig. 1) and $\hat{\text{EE}}_\infty$ is the asymptotic EE level reflecting limitations in uncertainty reduction. This model was fitted using fmincon in MATLAB (MathWorks, version 2019a) with a minimum mean square error method.

**Passive choices task (Exp. 2).** Each trial of the passive version of the task had two parts (Fig. 1c,d). First, the participants saw a configuration of dots on the screen (four purple dots and four white dots) that mapped onto an experimentally defined level of uncertainty, EE. On the basis of this configuration, the participants were required to indicate

how confident they were about the location of the hidden circle. They could do this by touching a rating scale on the side of the screen ranging between 0 and 100. A value of 0 on this scale meant that the participant had no idea where the hidden circle was, and 100 meant that the participant knew exactly where it was. Uncertainty estimation accuracy, a measure of how objective uncertainty is translated into subjective estimates, was defined as the slope of the relationship between EE and sign-flipped confidence ratings.

Next, once the participants had reported their confidence, the reward on offer appeared. They then were required to make 'Yes/No' decisions on the basis of whether they wished to place the blue disk given the reward on offer and the uncertainty they had just rated. There were four reward levels ($R$: 40, 65, 90 and 115 credits) and five uncertainty levels (EE: 16.3–24.4, 27.1–38.9, 57.5–58.9, 73.33–74.18 and 91.9–93.3 pixels). The participants were told that ten of the offers they accepted from the 100 trials would be randomly selected at the end of the experiment, and they would be required to play them (that is, localize the hidden circle using the blue disk). The credits they collected on these ten trials determined the monetary rewards they won in this version of the task. The score was calculated in the same way as in the training task. The participants were rewarded at a rate of €1 per 150 credits (Supplementary Table 25).

### Experimental paradigm for Exp. 3
A modified version of a well-validated effort-based decision-making task was used[41–46]. This task had a similar design as the Circle Quest passive choices task, but instead of uncertainty as the discounting attribute, the participants evaluated reward against physical effort (Fig. 2a). Reward in the task was represented as apples on trees, and effort levels were indicated by bars on the tree trunks. The higher the effort bar, the more effort the participants needed to exert to obtain the apples. Effort in the task was exerted by squeezing a hand-held dynamometer. The participants were first asked to squeeze the handle as hard as they could to measure their MVC. Crucially, the effort handle was calibrated on the basis of MVC for each participant. They were then familiarized with the different effort levels that they would encounter when making decisions. These effort levels corresponded to 16%, 32%, 48%, 64% and 80% of MVC. The participants experienced each effort level twice before performing the decision phase of the task, where they had to evaluate the worthiness of the reward (apples) against these effort levels. There were five reward levels corresponding to different numbers of apples (1, 4, 7, 10 and 13). The participants could indicate whether they wanted to accept or reject the offer on the screen by pressing either the left or right arrow on the keyboard to select 'Yes' or 'No', which were displayed on the sides of the screen. The positions of 'Yes' and 'No' changed randomly between trials. The participants were told that 10 of their decisions from the 125 trials would be randomly selected at the end of the experiment and that they would have to play them (that is, squeeze the effort handle to obtain the reward). They were rewarded on the basis of performance on these trials at a rate of €1 per ten apples. Before making their decisions, the participants had the chance to perform five practice decisions.

### Experimental paradigm for Exp. 4
In Exp. 4, a new version of Circle Quest (Exps. 1 and 2) was designed to investigate effort-based decision-making under uncertainty (Fig. 2). The participants were familiarized with uncertainty, reward and effort cues using the same training as in Exps. 1–3. This was done in three stages:

- Circle localization training. This was similar to what was done in Exps. 1 and 2.
- Confidence rating. Unlike in Exp. 2, confidence ratings were blocked and reported during the training phase rather than prior to each decision in the decision phase. This was done to ensure that offers with and without uncertainty were

experimentally matched during the decision phase. Ten catch trials were added to increase the range of uncertainty to fully capture subjective estimation of uncertainty as in Exp. 2. These included five trials at the lower range of uncertainty (EE < 23 pixels) and five at the higher range (EE > 91 pixels), resulting in an uncertainty range of EE between 17.9 and 93.3 pixels.

- Effort familiarization and calibration. This was the same as in Exp. 3.

After training, the participants were required to respond to (accept or reject) offers that had three attributes: reward, uncertainty and effort (Fig. 2a). Reward was presented as credits that the participants could win if they managed to complete two steps: (1) achieve the required level of effort (as in Exp. 3) and (2) find the location of the hidden circle without errors (as in Exps. 1 and 2). The blue disk used to localize the hidden circle appeared only when the required level of effort was achieved. If the effort level was met, the participants could then win credits depending on how far their localization using the blue disk was from the true location of the hidden purple circle. There were four levels of reward ($R$: 40, 65, 90 and 115 credits; the same as in Exp. 2), five levels of effort (16%, 32%, 48%, 64% and 80% of MVC; the same as in Exp. 3) and two levels of uncertainty (present and absent). The absence of uncertainty was indicated by showing the true location of the hidden circle on the screen when the offers were presented. On trials in which uncertainty was present, it corresponded to the midrange of EE used in Exp. 2 (31.8–73.95 pixels). This resulted in 40 different trial types (four reward levels × five effort levels × two uncertainty levels), and each was repeated five times over ten blocks (200 trials in total). Catch trials (ten in total, one in each uncertainty block) were also included in the decision phase. These were the catch trials used in the confidence rating phase that featured different uncertainty levels than what was otherwise used in the experiment. Each of the two levels of uncertainty (high and low) in these catch trials featured the five levels of effort. Such trials were designed to detect random responding in the task, especially when compared to general task performance.

The participants were told that at the end of the decision phase, 24 trials (12 with uncertainty) from the total 210 trials (including catch trials) would be randomly selected for them to play. Performance on these trials decided the reward that the participants eventually won. Similar to Exp. 2, they were rewarded at a rate of €1 per 150 credits (Supplementary Table 25).

### Statistical analyses of behavioural data

Statistical analyses and modelling were done in MATLAB R2019a or R version 4.0.2. Generalized mixed-effects models and robust regression models were fitted using the fitglm and fitlm functions in MATLAB, respectively. Bayesian mixed-effects modelling was performed using the Stan computational framework (http://mc-stan.org/) accessed using brms in R version 3.5.2 (ref. 103). Bayes factors were calculated using the brms hypothesis function. For group comparisons, we used Student's $t$-test if parametric assumptions were fulfilled and the Wilcoxon rank sum test if not. All statistical tests were two-tailed with a testing level ($\alpha$) of 0.05. The full description of the mixed-effects models used and the statistical results is reported in the Supplementary Information.

### MR data acquisition

MRI scans were obtained at the Acute Vascular Imaging Centre at John Radcliff Hospital (Oxford) using a SIEMENS Verio 3T scanner. High-resolution T1-weighted structural MR images (MPRAGE; 208 sagittal slices of 1 mm thickness; voxel size, 1 mm isotropic; TR/TE, 2,000/1.94 ms; flip angle, 8°; FOV read, 256; iPAT, 2; prescan-normalize) and T2-weighted fluid-attenuated inversion recovery images (192 sagittal slices of 1.05 mm thickness; voxel size, 1 × 1 × 1.1; TR/TE,

5,000/397 ms; FOV read, 256; iPAT, 2; partial Fourier, 7/8; fat saturation; prescan-normalize) were acquired. Four patients and two controls were not MRI compatible or did not consent to be scanned; therefore, imaging data were acquired for 15/19 patients and 17/19 controls. The average time intervals between MRI and behavioural testing for the ALE and control groups were 49.73 days (s.d. = 59.00) and 109.82 days (s.d. = 93.93), respectively.

### MR data analysis

VBM was performed to compare grey matter volumes between patients and controls using FSL-VBM[104] (http://fsl.fmrib.ox.ac.uk/fsl/fslwiki/FSLVBM). An optimized VBM protocol[105] was performed using FSL version 6.0 (ref. 106). Nonlinear registration was used to register structural brain-extracted and grey-matter-segmented images to the MNI 152 standard space. A study-specific grey matter template was then created using the resulting images. To avoid having a biased template, 15 controls were randomly selected to match the 15 patients who had MRI scans. Nonlinear registration was used to register all native grey matter images to the study-specific template, and this was modulated using the Jacobian of the warp field. These modulated grey matter images were then smoothed with an isotropic Gaussian kernel with a sigma of 4 mm. Finally, non-parametric testing using randomise with 5,000 permutations and corrected for multiple comparisons across space was used to detect voxel-wise differences in grey matter volumes between patients and controls.

Whole hippocampal volumes were extracted using T1 and T2 imaging in Freesurfer version 7.1 (http://surfer.nmr.mgh.harvard.edu/). The automated standard segmentation protocol was used. Hippocampal volumes were adjusted for intracranial volume (ICV) using the following equation[107,108]:

$$V_{\mathrm{adj}} = V - \beta \times (\mathrm{ICV} - \overline{\mathrm{ICV}}) \qquad (4)$$

where $V_{\mathrm{adj}}$ and $V$ are the adjusted and observed volumes, $\beta$ is the slope of the relationship between ICV and $V$ in a larger sample of healthy controls with similar demographics ($n = 31$, including the study sample and participants recruited for a different study), and $\overline{\mathrm{ICV}}$ is the mean ICV in this control sample. Amygdala volumes were also extracted and used as a control comparison region (Supplementary Fig. 5).

### Reporting summary

Further information on research design is available in the Nature Portfolio Reporting Summary linked to this article.

## Data availability

Anonymized participant data have been deposited on the Open Science Framework platform: https://osf.io/u4n2a/ (ref. 109).

## Code availability

The code for running the experiments and replicating the main results reported in the manuscript has been deposited on the Open Science Framework platform at https://osf.io/u4n2a/ (ref. 109).

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

## Acknowledgements

We thank K. Muhammed for helping in recruiting some of the patients for the study. This research was funded in whole, or in part, by the Wellcome Trust (grant no. 206330/Z/17/Z). For the purpose of open access, the authors have applied a CC BY public copyright licence to any Author Accepted Manuscript version arising from this submission. M.H. was funded by the Wellcome Trust. B.A. was funded by a Rhodes scholarship. S.G.M. was funded by an MRC Clinician Scientist Fellowship (no. MR/P00878/X). S.R.I. was funded by a senior clinical fellowship from the Medical Research Council (no. MR/V007173/1), a Wellcome Trust Fellowship (no. 104079/Z/14/Z), BMA Research Grants—Vera Down grant (2013) and Margaret Temple (2017), Epilepsy Research UK (P1201), the Fulbright UK-US commission (MS-Society research award) and the National Institute for Health Research Oxford Biomedical Research Centre. The funders had no role in study design, data collection and analysis, decision to publish or preparation of the manuscript.

## Author contributions

B.A., P.P. and M.H. designed the study. B.A., R.Z., S.R.I. and M.H. recruited the patients. B.A., S.T., M.R.M., A.G.-D. and R.Z. collected the data. B.A., P.P. and S.G.M. analysed the data. B.A. and M.H. wrote the paper.

## Competing interests

S.R.I. receives licensed royalties on patent application no. WO/2010/046716 entitled 'Neurological autoimmune disorders' and has filed two other patents entitled 'Diagnostic method and therapy' (nos WO2019211633 and US-2021-0071249-A1; PCT application no. WO202189788A1) and 'Biomarkers' (nos PCT/GB2022/050614 and WO202189788A1). The remaining authors declare no competing interests.

## Additional information

**Correspondence and requests for materials** should be addressed to Bahaaeddin Attaallah.

# Reporting Summary

## Statistics

For all statistical analyses, confirm that the following items are present in the figure legend, table legend, main text, or Methods section.

| n/a | Confirmed | |
|---|---|---|
| ☐ | ☒ | The exact sample size (*n*) for each experimental group/condition, given as a discrete number and unit of measurement |
| ☐ | ☒ | A statement on whether measurements were taken from distinct samples or whether the same sample was measured repeatedly |
| ☐ | ☒ | The statistical test(s) used AND whether they are one- or two-sided<br>*Only common tests should be described solely by name; describe more complex techniques in the Methods section.* |
| ☐ | ☒ | A description of all covariates tested |
| ☐ | ☒ | A description of any assumptions or corrections, such as tests of normality and adjustment for multiple comparisons |
| ☐ | ☒ | A full description of the statistical parameters including central tendency (e.g. means) or other basic estimates (e.g. regression coefficient) AND variation (e.g. standard deviation) or associated estimates of uncertainty (e.g. confidence intervals) |
| ☐ | ☒ | For null hypothesis testing, the test statistic (e.g. $F$, $t$, $r$) with confidence intervals, effect sizes, degrees of freedom and $P$ value noted<br>*Give P values as exact values whenever suitable.* |
| ☐ | ☒ | For Bayesian analysis, information on the choice of priors and Markov chain Monte Carlo settings |
| ☐ | ☒ | For hierarchical and complex designs, identification of the appropriate level for tests and full reporting of outcomes |
| ☐ | ☒ | Estimates of effect sizes (e.g. Cohen's *d*, Pearson's *r*), indicating how they were calculated |

*Our web collection on statistics for biologists contains articles on many of the points above.*

## Software and code

Policy information about availability of computer code

| | |
|---|---|
| Data collection | MATLAB (The MathWorks inc., version 2018b), Psychtoolbox v3. |
| Data analysis | MATLAB (The MathWorks inc., version 2019a), R version 3.5.2, Freesurfer version 7.1, FSL version 6.0. |

For manuscripts utilizing custom algorithms or software that are central to the research but not yet described in published literature, software must be made available to editors and reviewers. We strongly encourage code deposition in a community repository (e.g. GitHub). See the Nature Portfolio guidelines for submitting code & software for further information.

## Data

Policy information about availability of data

All manuscripts must include a data availability statement. This statement should provide the following information, where applicable:
- Accession codes, unique identifiers, or web links for publicly available datasets
- A description of any restrictions on data availability
- For clinical datasets or third party data, please ensure that the statement adheres to our policy

Anonymised participant data have been deposited on the Open Science Framework platform: https://osf.io/u4n2a/.

# Research involving human participants, their data, or biological material

Policy information about studies with <u>human participants or human data</u>. See also policy information about <u>sex, gender (identity/presentation), and sexual orientation</u> and <u>race, ethnicity and racism</u>.

| | |
|---|---|
| Reporting on sex and gender | Gender was determined based on self-reports.<br>Used only for purposes of matching patient and controls and as covariate where applicable. |
| Reporting on race, ethnicity, or other socially relevant groupings | N/A |
| Population characteristics | Exps. 1, 2 and 3:<br>N = 36 (19 ALE patients and 19 healthy Controls).<br>Age and gender: ALE (age: μ = 60.00, SD = ±11.36, 13 males). Controls (age: μ = 61.16, SD = ±11.71, 13 males).<br><br>In Exp. 4, N = 20 (eight ALE patients and 12 controls)<br>Age and gender: ALE (age: μ =55, SD = ±12.55, 6males). Controls (μ = 64.76, SD = ±7.42, 9 males)<br><br>Details characteristics of the two groups are described Tables in S1, S2, and S3 in Supplementary materials. |
| Recruitment | The ALE patient group was recruited based on a confirmed diagnosis through direct referrals from clinicians, whenever a patient with the condition was identified, in addition to the pre-existing pool of patients known to the cognitive neurology group in Oxford. All eligible patients identified during the data collection period were contacted and recruited upon their agreement to participate in the study. Age- and gender-matched controls were selected from a pool of volunteers who expressed interest in contributing to cognitive neurology research in Oxford. A computer code that matched age and gender and randomly selected eligible controls was utilized. Contact with potential candidates was established via phone and/or email. |
| Ethics oversight | University of Oxford ethics committee (RAS ID: 248379, Ethics Approval Reference: 18/SC/0448) |

Note that full information on the approval of the study protocol must also be provided in the manuscript.

# Field-specific reporting

Please select the one below that is the best fit for your research. If you are not sure, read the appropriate sections before making your selection.

☐ Life sciences   ☒ Behavioural & social sciences   ☐ Ecological, evolutionary & environmental sciences

For a reference copy of the document with all sections, see <u>nature.com/documents/nr-reporting-summary-flat.pdf</u>

# Behavioural & social sciences study design

All studies must disclose on these points even when the disclosure is negative.

| | |
|---|---|
| Study description | Quantitative case control lesion study. Data mainly from behavioural experiments. |
| Research sample | Autoimmune limbic encephalitis patients with LGI1/CASPRE2 anti-bodies and age- and gender matched controls. Demographics and group characteristics are summarized in Tables S1 and S2 of the manuscript. 19 individuals with a previously established diagnosis of ALE (age: μ = 60.00, SD = ±11.36, 13 males) were tested along with 19 healthy age- and gender-matched controls (age: μ = 61.16, SD = ±11.71, 13 males). In Exp. 4, eight ALE patients and 12 controls completed an additional follow-up task.<br><br>ALE patients are considered a lesion model for for focal hippocampal damage in humans, which is key to the question of the study investigating the role of the hippocampus in decision making. The study sample is representative of this group of patients with confirmed diagnosis and neuroimaging findings showing hippocampal atrophy. |
| Sampling strategy | Sampling relied on convenience and availability of patients as they present to the neurology clinic in Oxford or if they known patients from our database agreed to take part. Given the rarity of the condition, any patient in our data base who fitted the inclusion criteria was referred to take part in the study when it was running or invited to participate. Sample size was determined based on previous comparable work in ALE patients (Hanert et al., 2019; Spano et al., 2020a,b) as well as `previous research using the behavioural paradigms used in the study (Attaallah et al., 2022; Le Heron et al., 2018a,b; Petitet et al., 2021). See also description of recruitment above. |
| Data collection | All tasks were presented on a 17-inch touchscreen PC using MATLAB version 2018a and Psychtoolbox version 3. Participants sat in a quiet testing room, within reaching distance of the screen (about 50 cm). Questionnaire data were collected on an iPad tablet using RedCaP and Qualtrics.<br><br>An experimenter (sitting about 2-3 meters behind the participants) was present in the room at all time during behavioural testing. Their role was to explain the task at the beginning of the session (using an automated instruction script), and check that participants |

were engaged in the task throughout the session. They did not provide strategy advices when participants asked.

Researchers were not blind to the experimental condition or study hypothesis, but their influence on performance was abolished by the use of automated, computerised testing procedures.

MRI scans were obtained at Acute Vascular Imaging Centre (AVIC) at John Radcliff Hospital (Oxford) using SIEMENS Verio 3T scanner. Detailed description of the data collection process is explained in the manuscript.

| | |
|---|---|
| Timing | Experiments 1--3: Feb 2019- Feb 2020.<br>Experiment 4: Feb 2022  May 2022 (Post-covid interruption, especially collecting data from vulnerable patient group) |
| Data exclusions | No exclusions.<br>A number of rials from one of the patients for Exp. 2 were lost due to technical error. Data was still usable. |
| Non-participation | Exp.4:<br>11 patients: death (1), clinical deterioration (2), moved abroad (1), did not reply or declined (7).<br>7 controls: moved abroad (1), did not reply or declined (6). |
| Randomization | Participants were assigned to two groups: controls and cases. The cases were ALE patients. No randomisation is applicable across the two groups.<br>However, behavioural task trials were randomised and study blocks were counter-balanced across participants. |

# Reporting for specific materials, systems and methods

We require information from authors about some types of materials, experimental systems and methods used in many studies. Here, indicate whether each material, system or method listed is relevant to your study. If you are not sure if a list item applies to your research, read the appropriate section before selecting a response.

## Materials & experimental systems

| n/a | Involved in the study |
|---|---|
| ☒ | Antibodies |
| ☒ | Eukaryotic cell lines |
| ☒ | Palaeontology and archaeology |
| ☒ | Animals and other organisms |
| ☒ | Clinical data |
| ☒ | Dual use research of concern |
| ☒ | Plants |

## Methods

| n/a | Involved in the study |
|---|---|
| ☒ | ChIP-seq |
| ☒ | Flow cytometry |
| ☐ | ☒ MRI-based neuroimaging |

## Magnetic resonance imaging

### Experimental design

| | |
|---|---|
| Design type | N/A: Offline structural imaging. |
| Design specifications | N/A: no task-fMRI data collected |
| Behavioral performance measures | N/A: Reward and uncertainty sensitivity from Exp. 2 from offline behavioural tasks (not task-fMRI). |

### Acquisition

| | |
|---|---|
| Imaging type(s) | Structural |
| Field strength | 3T |
| Sequence & imaging parameters | r. High-resolution T1-weighted structural MR images (MPRAGE; 208 sagittal slices of 1 mm thickness, voxel size = 1 mm isotropic, TR/TE = 2000/1.94 ms; flip angle = 8∘, FOV read = 256, iPAT =2, prescan-normalise) and T2 weighted fluid attenuated inversion recovery (FLAIR) images (192 sagittal slices of 1.05 mm thickness, voxel size = 1×1×1.1, TR/TE = 5000/397 ms; FOV read = 256; iPAT = 2, partial Fourier = 7/8, fat saturation, prescan-normalise) were acquired. |
| Area of acquisition | Whole brain. |
| Diffusion MRI | ☐ Used    ☒ Not used |

### Preprocessing

| | |
|---|---|
| Preprocessing software | FSL 6.0 and FreeSurfer 7.1. |

| | |
|---|---|
| Normalization | Non-linear registration was used to register structural brain-extracted and grey matter-segmented images to the MNI 152 standard space. |
| Normalization template | MNI 152 |
| Noise and artifact removal | N/A: Structural imaging |
| Volume censoring | N/A: Structural imaging |

## Statistical modeling & inference

| | |
|---|---|
| Model type and settings | N/A: Structural analysis with GLM investigating VBM group differences. |
| Effect(s) tested | N/A |
| Specify type of analysis: | ☒ Whole brain ☐ ROI-based ☐ Both |
| Statistic type for inference<br>(See Eklund et al. 2016) | VBM (voxel-wise). |
| Correction | Non-parametric testing using randomise in FSL with 5000 permutations and corrected for multiple comparisons across space was used to detect voxel-wise differences in grey matter volumes between patients and controls. |

## Models & analysis

| n/a | Involved in the study |
|---|---|
| ☒ ☐ | Functional and/or effective connectivity |
| ☒ ☐ | Graph analysis |
| ☒ ☐ | Multivariate modeling or predictive analysis |

