## [Peer Review File · Nature Human Behaviour]

Peer Review Information

Journal: Nature Human Behaviour

Manuscript Title: Role of the hippocampus in decision making under uncertainty

Corresponding author name(s): Bahaaeddin Attaallah

Reviewer Comments & Decisions:

Decision Letter, initial version:

15th September 2023

Dear Dr Attaallah,

Thank you once again for your manuscript, entitled "Role of the hippocampus in decision making under uncertainty", and for your patience during the peer review process.

Your Article has now been evaluated by 3 referees. You will see from their comments copied below that, although they find your work of potential interest, they have raised quite substantial concerns. In light of these comments, we cannot accept the manuscript for publication, but would be interested in considering a revised version if you are willing and able to fully address reviewer and editorial concerns.

We hope you will find the referees' comments useful as you decide how to proceed. If you wish to submit a substantially revised manuscript, please bear in mind that we will be reluctant to approach the referees again in the absence of major revisions. We are committed to providing a fair and constructive peer-review process. Do not hesitate to contact us if there are specific requests from the reviewers that you believe are technically impossible or unlikely to yield a meaningful outcome.

In particular, we ask you to thoroughly address the following issues:

- 1) provide convincing evidence that there are no memory confounds in the decision making effects (R2 point 1)
- 2) substantially improve the logical flow and presentation of the results (R1 points 1 and 2)
- 3) provide additional clinical details about the ALE patients (R3 points 1 and 2)

Finally, your revised manuscript must comply fully with our editorial policies and formatting requirements. Failure to do so will result in your manuscript being returned to you, which will delay its consideration. To assist you in this process, I have attached a checklist that lists all of our requirements. If you have any questions about any of our policies or formatting, please don't hesitate

to contact me.

If you wish to submit a suitably revised manuscript, we would hope to receive it within 4 months. I would be grateful if you could contact us as soon as possible if you foresee difficulties with meeting this target resubmission date.

- Include a "Response to the editors and reviewers" document detailing, point-by-point, how you addressed each editor and referee comment. If no action was taken to address a point, you must provide a compelling argument. When formatting this document, please respond to each reviewer comment individually, including the full text of the reviewer comment verbatim followed by your response to the individual point. This response will be used by the editors to evaluate your revision and sent back to the reviewers along with the revised manuscript.
- Highlight all changes made to your manuscript or provide us with a version that tracks changes.

[REDACTED]

Thank you for the opportunity to review your work. Please do not hesitate to contact me if you have any questions or would like to discuss the required revisions further.

Sincerely,

Giacomo Ariani
Editor
Nature Human Behaviour

Reviewer expertise:

Reviewer #1: decision making, uncertainty, reward, memory

Reviewer #2: hippocampus role, lesions/deficits

Reviewer #3: autoimmune limbic encephalitis, hippocampus

REVIEWER COMMENTS:

Reviewer #1:

Remarks to the Author:

The authors examine decision-making in conditions with varying uncertainty, reward value, and effort, in patients with selective hippocampal damage (ALE, acute limbic encephalitis) and matched healthy controls. The role of the hippocampus in decision-making is a very active area of research, with many open interesting questions, so the current results are potentially of wide interest. The results of interest are primarily driven by decisions to accept or reject offers to "play out" offers with 1) uncertainty and varying reward, and 2) relative certainty and varying effort. The ALE patients were less sensitive to reward but showed intact sensitivity to uncertainty itself, while in decisions of relative certainty and varying effort, no overall differences were found. The authors suggest that hippocampal damage affects processing of value-related attributes when there is uncertainty. The authors also integrate a final experiment in a small subgroup with somewhat consistent findings. Finally, this interpretation may also be supported by an "active" experimental condition where ALE patients acted more quickly and more often to try to resolve uncertainty.

Initially, I found it initially difficult to fit these findings into a clear conclusion. However, after further consideration and thoughts about how the results could be presented differently, I believe there is an interesting and relatively clear story here about the role of the hippocampus in decision-making under uncertainty, which would have wide potential interest. My comments below are primarily suggestions about how to improve the presentation of the results, which I think in this case can substantially affect the perceived strength of the paper.

1. The presentation could be improved already in the results of Exp 1. (As a side-note, I greatly appreciated the summary of all the different experiments at the start of the Results section.)

The initial result of Exp 1 regarding the number of samples in different conditions was confusing and not intuitive, and it took a lot of time to process. As the main story doesn't really start until Exp 2, it would help a lot to make Exp 1 results clear and limited. Perhaps start with the optimal sampling model first, otherwise readers may get hung-up on trying to understand a $p = 0.038$ interaction effect in an unfamiliar task. (The discussion of the interaction being stronger in patients than controls was also confusing.) Further, as the authors really only seem to mention this experiment in the Discussion section in terms of the speed of sampling, I would suggest including those results in a main figure. Perhaps to make the initial results easy to follow, and to preview what is in the Discussion, discuss and display something like Fig 5 a and d on optimal sampling, plus Fig S2 panel c and d for speed of samples?

2. The findings in Exp 2 and 3 are clear, and support the overall story. However, things get more difficult again in the presentation of Exp 4, where a subset of participants returned (only 8 of 19 patients).

Based on what came before, what we are looking for here are 3-way interactions, between ALE patients x reward x uncertainty, and ALE patients x effort x uncertainty. It would make sense to start with those analyses only – especially given the low number of participants, which should limit additional analyses that do not inform a priori predictions. (Similarly, in the plots in Fig. 8, it seems like the lines should be plotted separately for the certain and uncertain conditions.)

However, the Exp 4 results start out collapsing across uncertainty, with patient x reward and patient x

effort effect, finding that ALE patients were less sensitive to reward or effort overall. That undermines the selective uncertainty-related story, as this result implies (whether or not that is true) that patients are also less sensitive to valuation-related variables even in conditions of certainty.

If the authors present the full interactions first, it would provide clearer support. The 3-way interaction including reward (ALE x reward x uncertainty) is significant. It would help to highlight here that this finding is not just a same-patient replication of Exp 2, as Exp 4 included a new certain condition.

However, the 3-way interaction including effort is not significant. It does appear to be numerically in the expected direction, and these statistics should be reported in the Results. The failure to find an effect should be discussed, in the context of the obvious limitations of only having a dataset limited to 8 patients.

As a side-note, I'm curious about the certain condition, and whether the patients showed reduced sensitivity to reward or effort in this condition as well, but I think such exploratory analyses should be limited given the low number of subjects.

3. The attempt to link Exp 2 and Exp 1 with an across-participant behavior-behavior correlation with a low number of participants ($n = 19$) is not a strong approach. Perhaps just note this as an exploratory, underpowered, analysis?

(In contrast, the later behavior-anatomy correlation rests on a more solid statistical foundation, given the likely very high test-retest reliability in the anatomy component of that analysis.)

4. For the brain results, please add stats for the equivalent correlation between uncertainty and volume (and if reward and uncertainty correlations are different).

Minor points:

5. First sentence of abstract, suggest perhaps rephrasing in a positive way (e.g. "beginning to be more understood"), or at least replace "poorly" with "little"?

6. p. 5 last sentence, suggest rephrasing e.g. "are an important additional step"?

7. p. 8, Exp 2 please note what fraction of trials are selected out of the total (e.g. 10 out of 200). Also include this detail for Exp 3 and Exp 4.

8. p. 11, the description of Exp 4 does not follow the same organization as the earlier experiments – for example, it is missing the statement that n trials would be selected for playing for real.

9. Fig 4 caption: the phrase "Once training was completed" sticks out, because it isn't described until later. Perhaps add "see below".

10. Fig 4 caption: please clarify with statement that trials were selected from the "Yes" (accepted) trials.

11. Results: for each section, please clearly indicate the experiment being discussed when that section

starts (e.g. on p. 15, it isn't clear immediately that the experiment under discussion has shifted to Exp 2).

12. p. 15, potential typo " $z = 3.79, p = 0.70$ " – these numbers do not make sense together.

13. Figure 5e title, suggest using word "credits", as the abbreviation "cfs" isn't helpful and it is not defined.

14. Figure 6a and similar plots: suggest indicating individual subjects with light-color lines.

15. Experiment 4, p. 20 please add a mention of the actual (smaller) number of participants in each group somewhere.

16. Figure 9, suggest note that the behavioral data comes from Exp 2.

17. Discussion, p. 28, regarding hippocampus and entropy, Bornstein and Daw, 2013, PLoS CB would be a good citation to add.

18. Discussion, end of p. 30, Wimmer et al. 2023 PNAS would be a recent citation in human MEG to add.

19. Table S3: note reference group, as done in S7.

Reviewer #2:

Remarks to the Author:

This manuscript examines the role of the hippocampus in decision making in situations of uncertainty. It proceeds by comparing the performance of patients with autoimmune limbic encephalitis (ALE), a condition that primarily yields lesion of the hippocampus, to that of controls across four experiments featuring decision trials varying in reward amount, required effort, and uncertainty. The hippocampus is a brain region known to be dedicated to memory, and its role in decision making still needs to be elucidated: the results of the current paper contribute nicely to this area of research and should be of interest to the scientific community. Considering the rareness of ALE, the sample size (N=19 ALE patients and N=19 controls) was quite nice. The experiments and statistical analyses were extensive and appeared to be well conducted, and the open availability of the data and code was appreciated. There were some concerns related to being able to infer that the impairments that were revealed could indeed be attributed to a role of the hippocampus in uncertainty, rather than to memory confounds or to other roles of the hippocampus. Detailed comments related to these concerns are provided below for the authors to consider.

(1) Addressing possible memory confounds. One core difficulty in research examining the role of the hippocampus in decision making is ensuring that there is no memory confound, with concern for example that patients may partially forget instructions mid-task and get confused, resulting in impaired decision making. The authors attended to these concerns by making the tasks very visual and keeping some visual cues on the screen during the trials. Nonetheless, the tasks were quite

complex, and the full instructions were not displayed on the screen at all times. How did the authors ensure that impairment was not related to confusion or to some memory failure when the tasks became more complex, especially when participants carried out up to 200 decisions in a row without the consequences being played out?

A lack of difference in response reaction time was mentioned as a way to indicate that the trials were not experienced as generally “more difficult” for the patients. But random responding could nonetheless be carried out quickly. The inclusion of catch trials would have been more effective for checking that patients were still on task.

There was some mention of catch trials in the task related to estimating uncertainty (c.f. caption of Figure 4 and text on page 41). These catch trials were not detailed in the method or results, though, and more detail is warranted to clarify what they consisted of and how the patients fared compared to the controls. But more importantly, were catch trials also included during the decision trials of interest to ensure lack of random responding? And if so, what did they look like and how did the patients fare compare to the controls? No matter what, the topic of a potential memory-based or confusion-based confound should probably be thoroughly discussed in the discussion section as a potential limitation for the present conclusions.

(2) Implications/Discussion of findings. Some of the implications of the present findings and comparison with the literature could be strengthened in the Discussion section.

2.a. First, I am not sure that I followed the direct link that was made between episodic mental projection of future events and intolerance of uncertainty (pages 27-28). These sections could benefit from clarifications.

2.b. It was also not clear to me that findings of impairments when reward & uncertainty demands were present but not when reward & effort demands were present do not simply reflect that the task with uncertainty demands was more complex than the task with effort demands. Reaction time was compared across participant groups. But was reaction time compared across tasks?

2.c. The present results are interestingly quite consistent with previous studies that have reported an impairment in sensitivity to change in one task feature when decisions required integrating or updating several task components. For example, in the intertemporal choice animal hippocampal lesion literature, Bett et al. 2015 (Hippocampus, DOI 10.1002/hipo.22400) demonstrated diminished sensitivity to change in the rewards, and Masuda et al. 2020 (eLife, DOI:

<https://doi.org/10.7554/eLife.52466>) demonstrated diminished sensitivity to changes in the delay.

Similarly, in a more recent human study on patients with hippocampal lesions, Patt et al. 2023 (Journal of Neuroscience, DOI: <https://doi.org/10.1523/JNEUROSCI.2250-22.2023>) demonstrated diminished sensitivity to changes in the delay during experiential intertemporal choice.

These similar patterns of findings could be explained by a role of the hippocampus in processes other than uncertainty, for example from the role of the hippocampus in constructing representations that require integrating several complex task features (Yonelinas 2013, Beh Brain Research, 254, p.34) or from a role of the hippocampus in experimental range adaptation (Cox & Kable, 2014, 34, p.16533 J. Neuroscience). Some of these considerations could be added in the Discussion section.

Reviewer #3:

Remarks to the Author:

In their manuscript entitled “Role of the hippocampus in decision making under uncertainty” Attaallah et al. hypothesize that the human hippocampus may play a specific role in decision making involving

evaluation of uncertain values.

They studied a group of 19 individuals with acute autoimmune limbic encephalitis with anti-LGI1 and/or CASPR2 antibodies deemed to focally affect the hippocampus compared to 19 healthy age- and gender-matched controls on how they evaluate reward against uncertainty compared to reward against physical effort using four experiments ((i) active information gathering prior to committing to decisions under uncertainty using the active Circle Quest paradigm, (ii) passive decision making under uncertainty using the passive Circle Quest paradigm, (iii) effort-based decision making, and (iv) effort-based decision making under uncertainty) requiring participants to make trade-offs between reward, uncertainty and effort.

They obtained the following major findings:

1. ALE patients demonstrated blunted sensitivity to reward and effort whenever uncertainty was considered, despite demonstrating intact uncertainty sensitivity.
2. By contrast, valuation of reward and effort was intact on uncertainty-free tasks.
3. Reduced sensitivity to changes in reward under uncertainty correlated with severity of hippocampal damage.

Authors conclude a context-sensitive role of the hippocampus in value-based decision making specifically under conditions of uncertainty.

This is a very well designed and conducted study demonstrating for the role of the hippocampus in decision making and extend our knowledge regarding cognitive/behavioural deficits in ALE patients.

There are some minor issues that should be addressed by the authors:

Clinical details regarding the ALE patients are warranted. Especially, disease duration and treatments need to be described.

How were ALE and HC groups matched regarding education?

What were the time intervals between behavioural testing and MRI in both groups?

Why did the authors use absolute values for the hippocampal volume instead of normalizing them to the brain/head volume?

Author Rebuttal to Initial comments

Reviewer #1:

Remarks to the Author:

The authors examine decision-making in conditions with varying uncertainty, reward value, and effort, in patients with selective hippocampal damage (ALE, acute limbic encephalitis) and matched healthy controls. The role of the hippocampus in decision-making is a very active area of research, with many open interesting questions, so the current results are potentially of wide interest. The results of interest are primarily driven by decisions to accept or reject offers to "play out" offers with 1) uncertainty and varying reward, and 2) relative certainty and varying effort. The ALE patients were less sensitive to reward but showed intact sensitivity to uncertainty itself, while in decisions of relative certainty and varying effort, no overall differences were found. The authors suggest that hippocampal damage affects processing of value-related attributes when there is uncertainty. The authors also integrate a final experiment in a small subgroup with somewhat consistent findings. Finally, this interpretation may also be supported by an "active" experimental condition where ALE patients acted more quickly and more often to try to resolve uncertainty.

Initially, I found it initially difficult to fit these findings into a clear conclusion. However, after further consideration and thoughts about how the results could be presented differently, I believe there is an interesting and relatively clear story here about the role of the hippocampus in decision-making under uncertainty, which would have wide potential interest. My comments below are primarily suggestions about how to improve the presentation of the results, which I think in this case can substantially affect the perceived strength of the paper.

1. The presentation could be improved already in the results of Exp 1. (As a side-note, I greatly appreciated the summary of all the different experiments at the start of the Results section.)

The initial result of Exp 1 regarding the number of samples in different conditions was confusing and not intuitive, and it took a lot of time to process. As the main story doesn't really start until Exp 2, it would help a lot to make Exp 1 results clear and limited.

Perhaps start with the optimal sampling model first, otherwise readers may get hung-up on trying to understand a $p = 0.038$ interaction effect in an unfamiliar task. (The discussion of the interaction being stronger in patients than controls was also confusing.) Further, as the authors really only seem to mention this experiment in the Discussion section in terms of the speed of sampling, I would suggest including those results in a main figure. Perhaps to make the initial results easy to follow, and to preview what is in the Discussion, discuss and display something like Fig 5 a and d on optimal sampling, plus Fig S2 panel c and d for speed of samples?

We thank the reviewer for this suggestion. We have now reviewed the Results subsection of Exp. 1, taking into consideration their helpful comments.

The section now starts with a brief summary of important raw results including how participants responded to experimental manipulations, mainly sampling cost and initial credit reserve as well as the trajectory that uncertainty follows as a function of sampling. We think these initial results are important to provide evidence that participants understood the task and help introduce results of optimal sampling analysis.

As suggested, we limit subsequent analyses to discuss deviation from optimality and differences in sampling speed. The explanation of the three-way interaction was also reviewed for more clarity. Figure 5 has also been edited accordingly.

“Exp. 1: Reduced sensitivity to changes in information cost in ALE patients

In the active sampling task (Exp. 1), participants in both groups acquired samples to reduce uncertainty. Similar to previous reports (Attaallah et al., 2022; Petitet et al., 2021), reduction in uncertainty followed an exponential decay as a function of the number of samples acquired, indicating purposeful sampling abiding with task rules (Figure 5a.). Both patients and healthy controls behaved rationally, sampling less when acquiring samples was more expensive (Main effect of η_s on the number of samples acquired: $\beta = -0.11$, $t_{2272} = -6.25$, $p < 0.0001$, Table S4), and not responding to changes in initial reward reserve (Main effect of R_0 on the number of samples acquired: $\beta = 0.023$, $t_{2272} = 1.29$, $p = 0.20$). Patients' and controls' performance in the active task was evaluated with regard to optimal sampling behaviour to determine whether they tended to under- or over-sample (Figure 5b.). Optimal sampling refers to the number of samples, s^* , that maximises expected return, given the current cost-benefit structure (R_0 , η_s , η_e) and search efficiency (i.e., the rate at which participants reduce uncertainty from one sample to the next, parameterised as the information extraction rate, α , see Materials and Methods). Both ALE patients and healthy controls over-sampled when sampling cost was high (Deviation from optimal; ALE: $\beta = 3.93$, $t_{1138} = 4.32$, $p < 0.0001$; Controls: $\beta = 1.93$, $t_{1138} = 3.485$, $p < 0.001$, see also Table S5), but patients over-sampled to a greater extent than controls when the initial reward reserve was high in these conditions ($z = 2.267$, $p = 0.023$, Figure 5b.). There was no significant difference between the two groups when the sampling cost was low ($t_{36} = 0.579$, $p = 0.56$). ALE patients' sampling speed was also less deterred by increasing sampling cost, compared to controls (ALE $\times \eta_s$: $\beta = -0.0719$, $t_{2272} = -2.14$, $p = 0.033$, Figure 5c., Table S4). This suggests faster and less deliberate sampling in ALE when sampling costs are high, as seen in a speed efficiency-trade-off characterising sampling behaviour in both groups ($ISI \sim \alpha$; ALE: $\beta = 0.26$, $t_{1138} = 4.82$, $p < 0.0001$, Controls: $\beta = 0.137$, $t_{1138} = 2.96$, $p = 0.0032$, Figure 5d., Table S6). These findings suggest that ALE patients' sampling behaviour demonstrates, at least partially, blunted sensitivity to sampling cost leading to over-sampling (i.e., giving up more reward than needed in exchange for information) as well as faster sampling (see Supplementary materials for a computational model characterising this difference in sensitivity to sampling cost, Figure S1). This had consequences in terms of total reward received as patients' scores suffered to a greater extent than controls' when the sampling cost increased (ALE $\times \eta_s$: $\beta = -3.66$, $t_{2272} = -2.00$, $p = 0.046$, Table S4).

Figure 5: Exp. 1 - Reduced sensitivity to changes in information cost in ALE patients. **a.** Uncertainty (indexed as expected error, EE) decreases with sampling and follows an exponential decay slope on average in both patients and controls. **b.** ALE patients and healthy controls over-sampled when sampling cost was high. Patients, however, over-sampled to a greater extent than controls, mainly when initial reward reserve and sampling cost both increased. There was no significant difference between the two groups at low-cost conditions. **c.** Higher sampling cost was associated with slower sampling rates. This effect was less evident in ALE patients compared to controls, resulting in faster sampling in ALE patients in high sampling cost conditions. **d.** Sampling behaviour was characterised by a speed-efficiency trade-off whereby faster sampling rates (shorter ISI) were associated with lower sampling efficiency (smaller α). The figure shows this trade-off for the high sampling cost conditions, in which ALE patients sampled faster and also over-sampled when initial reward was high. Error bars show SEM. *: $p < 0.05$. See Tables S4, S5 & S6 for full statistical details.

2. The findings in Exp 2 and 3 are clear, and support the overall story. However, things get more difficult again in the presentation of Exp 4, where a subset of participants returned (only 8 of 19 patients).

Based on what came before, what we are looking for here are 3-way interactions, between ALE patients x reward x uncertainty, and ALE patients x effort x uncertainty. It would make sense to start with those analyses only – especially given the low number of participants, which should limit additional analyses that do not inform a priori predictions. (Similarly, in the plots in Fig. 8, it seems like the lines should be plotted separately for the certain and uncertain conditions.)

However, the Exp 4 results start out collapsing across uncertainty, with patient x reward and patient x effort effect, finding that ALE patients were less sensitive to reward or effort overall. That undermines the selective uncertainty-related story, as this result implies (whether or not that is true) that patients are also less sensitive to valuation-related variables even in conditions of certainty.

If the authors present the full interactions first, it would provide clearer support. The 3-way interaction including reward (ALE x reward x uncertainty) is significant. It would help to highlight here that this finding is not just a same-patient replication of Exp 2, as Exp 4 included a new certain condition.

However, the 3-way interaction including effort is not significant. It does appear to be numerically in the expected direction, and these statistics should be reported in the Results. The failure to find an effect should be discussed, in the context of the obvious limitations of only having a dataset limited to 8 patients.

We have edited this result subsection of Exp. 4 to focus on three-way interactions and their implications along with the limitation due to sample size. While reviewing our analysis, we discovered a mistake in encoding the effort component of the model; it was differently coded for fixed and random effects. This error has now been corrected and we have updated the manuscript with this. This model has a better fit (lower AIC, compared to the one with the error) and reveals a marginally significant 3-way interaction including effort. We now also highlight that the results from this experiment are not simply a replication of Exp. 2. Figure 8 has been modified to show separate lines for certain and uncertain conditions.

“The results showed that indeed ALE patients were less sensitive to reward than controls when making effort-based decisions under uncertain conditions (Reward × uncertainty × ALE: $\beta = +0.684$, $t_{4184} = 2.37$, $p = 0.018$). The three-way interaction investigating group difference in effort sensitivity under uncertainty was also marginally significant (Reward × uncertainty × ALE: $\beta = -0.541$, $t_{4184} = -1.97$, $p = 0.049$). There was no significant difference between the two groups in their sensitivity to uncertainty (ALE × EE : $\beta = 1.09$, $t_{4184} = 1.25$, $p = 0.21$).”

“ Importantly, this is not a replication of Exp. 2 results because i) decision-making involves effort consideration in addition to uncertainty and reward and ii) the task includes a new certain condition against which the effect

of uncertainty is compared. That said, these findings from Exp. 4 should be interpreted with caution due to the small sample size. “

Figure 8: **Exp. 4 - Blunted reward and effort sensitivity under uncertainty in ALE patients.** In Exp. 4, participants were required to accept/reject offers taking into consideration three attributes: reward, physical effort, and uncertainty. The results showed that ALE patients, compared to controls, were less sensitive to changes in reward under uncertainty and effort (the slope difference between the continuous and dashed lines across the two groups) while having intact uncertainty sensitivity (degree of downward shift between continued and dashed lines). Such results are consistent with findings from Exps. 1 & 2, highlighting disrupted reward and cost valuation in ALE patients under uncertainty. The shadow around lines shows \pm SEM. See Table S14 for full statistical details.

As a side-note, I'm curious about the certain condition, and whether the patients showed reduced sensitivity to reward or effort in this condition as well, but I think such exploratory analyses should be limited given the low number of subjects.

As correctly noted, performing this analysis should be limited due to sample size. However, we have done so to explore the results further as we do agree it is an interesting aspect to consider. The results showed reduced sensitivity to both reward and effort in the condition where uncertainty was absent (ALExreward: $\beta = -1.70$, $t(2092) = -3.43$, $p < 0.001$, <0.001 & ALExEffort: $\beta = 2.01$, $t(2092) = 3.89$, $p < 0.001$). If such a result is validated with a larger sample size, it can open another door to the discussion regarding how uncertainty is considered in these decision-making processes. This might point to a meta-cognitive aspect of the computation of subjective

value under uncertainty which not only takes into consideration the degree of uncertainty but also whether uncertainty absence should be actively considered. The absence of uncertainty might not be equal to certainty, i.e., being consciously aware of the absence of uncertainty (Exp. 4 when uncertainty is absent) is not the same as making a decision without actively considering whether it is present or absent (Exps. 2 & 3). This meta-cognitive cognitive aspect might be one of the main differences between Exp. 3 & the certain conditions in Exp. 4. We think this can be an interesting area for future research.

3. The attempt to link Exp 2 and Exp 1 with an across-participant behavior-behavior correlation with a low number of participants (n = 19) is not a strong approach. Perhaps just note this as an exploratory, underpowered, analysis?

(In contrast, the later behavior-anatomy correlation rests on a more solid statistical foundation, given the likely very high test-retest reliability in the anatomy component of that analysis.)

We have mentioned that this is an exploratory analysis and down-toned the significance of these results.

4. For the brain results, please add stats for the equivalent correlation between uncertainty and volume (and if reward and uncertainty correlations are different).

We have added the stats as requested to the main findings as follows:

“On the other hand, the correlation between uncertainty sensitivity and total hippocampal volumes was not significant (Model R² = 0.062, $t_{13} = 0.92$, $p = 0.37$).”

Minor points:

5. First sentence of abstract, suggest perhaps rephrasing in a positive way (e.g. "beginning to be more understood"), or at least replace "poorly" with "little"?

We thank the reviewer for their suggestion. We have rephrased the sentence as suggested.

"The role of the hippocampus is beginning to be more understood"

6. p. 5 last sentence, suggest rephrasing e.g. "are an important additional step"?

We thank the reviewer for their suggestion. We have rephrased the sentence as suggested.

"The findings might represent an important additional step in understanding selective hippocampal contributions to goal-directed and motivated behaviour."

7. p. 8, Exp 2 please note what fraction of trials are selected out of the total (e.g. 10 out of 200). Also include this detail for Exp 3 and Exp 4.

We have now added the requested information. In summary, participants played 10 out of 100 trials in Exp. 2, 10 out of 125 trials in Exp. 3, and 24 out of 210 trials (10 of which were catch trials).

8. p. 11, the description of Exp 4 does not follow the same organization as the earlier experiments – for example, it is missing the statement that n trials would be selected for playing for real.

We have rewritten this subsection in the Results and Methods sections to make it more consistent with the previous structure used to describe earlier experiments.

From the Results section:

"

Exp. 4 – Effort-based decision making under uncertainty

In the fourth experiment (Exp. 4), participants were re-invited to perform a third version of the Circle Quest paradigm but this time designed to investigate effort-based decision-making under uncertainty. The task was similar to those in Exps. 2 and 3. However, instead of making decisions (accept/reject) based on two attributes (reward vs. uncertainty as in Exp. 2 and reward vs. effort in Exp. 3), participants now had to make decisions under the three attributes together (reward, uncertainty and effort simultaneously) (Figure 4a.). Before

engaging in the decision making phase, participants were familiarised with the task using an interactive tutorial and then trained on circle localisation (similar to Exps. 1 & 2) and effort practice (similar to Exp. 3). Unlike Exp. 2, where participants estimated uncertainty prior to each decision, participants in this task gave these reports in one block before the decision phase. This accommodated two uncertainty conditions: present or absent, ensuring an experimentally balanced design for decision trials across these two conditions. On each trial in the decision phase, participants responded to offers considering three attributes: • Reward, depicted as credits with the same four levels as Exp. 2. • Effort, represented by the bar height on a rectangle, mirroring the five levels from Exp. 3. Successfully achieving the effort level unveiled the blue disc used in localising the hidden circle to win credits. • Uncertainty, showcased through dot configurations (as in Exps. 1 & 2), was absent on half of the trials by revealing the true circle location. When present, uncertainty ranged between expected errors (EE) of 31.8-73.95 pixels, similar to Exp. 2. Thus, similar to Exp. 2, each trial presented participants with credit offers attainable upon accurately localising the hidden circle. However, achieving the effort level designated for the trial was essential to reveal the localisation disc. Participants had to make 'Yes/No' decisions across 200 trials, encompassing three economic attributes: uncertainty (present or absent), reward (four levels), and effort (five levels). Ten additional catch trials were included when uncertainty was present, expanding the range of uncertainty beyond that used in the main task. Participants were informed that 24 'Yes' decisions would be randomly selected at the task's end, providing them an opportunity to play and receive rewards."

From the Materials and Methods section:

"Experimental paradigm – Exp. 4

In Exp. 4, a novel version of Circle Quest (Exps. 1 & 2) was designed to investigate effort-based decision making under uncertainty (Figure 4). Participants were familiarised with uncertainty, reward and effort cues using the same training as in Exps. 1–3. This was done in three stages:

- **Circle localisation training** similar to what was done in Exp. 1 & 2.
- **Confidence rating.** Unlike Exp. 2, confidence ratings were blocked and reported during the training phase rather than prior to each decision in the decision phase. This was done to ensure that offers with and without uncertainty were experimentally matched during the decision phase. Ten catch trials were added to increase the range of uncertainty to fully capture subjective estimation of uncertainty as in Exp. 2. These included five trials at the lower range of uncertainty ($EE < 23$ pixels) and five at the higher range ($EE > 91$ pixels) resulting in uncertainty range of EE between 17.9 and 93.3 pixels.
- **Effort familiarisation and calibration.** This was the same as in Exp. 3.

After training, participants were required to respond to (accept or reject) offers that had three attributes: reward, uncertainty and effort (Figure 4a.). Reward was presented as credits that participants could win if they managed to complete two steps: i) achieve the required level of effort (as in Exp. 3) and ii) find the location of the hidden circle without errors (as in Exps. 1 & 2). The blue disc used to localise the hidden circle appeared only when the required level of effort was achieved. If effort level was met, participants then could win credits depending on how far their localisation using the blue disc was from the true location of the hidden purple circle. There were four levels of reward (R: 40, 65, 90, 115 credits; same as in Exp. 2), five levels of effort (16, 32, 48, 64 and 0.8% of MVC; same as in Exp. 3) and two levels of uncertainty (present and absent). Absence of

uncertainty was indicated by showing the true location of the hidden circle on the screen when offers were presented. On trials in which uncertainty was present, it corresponded to the mid-range of expected error (EE) used in Exp. 2 (EE: 31.8-73.95 pixels). This resulted in 40 different trial types (four reward levels × five effort levels × two uncertainty levels), and each was repeated five times over ten blocks (200 trials in total). Catch trials (10 in total, one in each block) were also included in the decision phase. These were the catch trials used in the confidence rating phase that featured different uncertainty levels than what was otherwise used in the experiment. Each of the two levels of uncertainty (high and low) in these catch trials featured the five levels of effort. Such trials were designed to detect random responding in the task, especially when compared to general task performance. Participants were told that at the end of the decision phase, 24 trials (12 with uncertainty) from the total 210 trials (including catch trials) will be randomly selected for them to play. Performance on these trials decided the reward that participants eventually won. Similar to Exp. 2, they were rewarded at a rate of £1 per 150 credits (Table S24)."

9. Fig 4 caption: the phrase "Once training was completed" sticks out, because it isn't described until later. Perhaps add "see below".

Thanks for pointing this out. We have now added 'see below' at the end of the phrase.

10. Fig 4 caption: please clarify with statement that trials were selected from the "Yes" (accepted) trials.

This was added.

11. Results: for each section, please clearly indicate the experiment being discussed when that section starts (e.g. on p. 15, it isn't clear immediately that the experiment under discussion has shifted to Exp 2).

We have now edited to Results section headings to include the experiment being discussed as well as mentioning the experiment being discussed in the initial sentence for each subsection.

12. p. 15, potential typo " $z = 3.79$, $p = 0.70$ " – these numbers do not make sense together.

This was indeed a typo. " $z = -0.379$ ". This is now corrected. Thanks for pointing this out.

13. Figure 5e title, suggest using word "credits", as the abbreviation "cnds" isn't helpful and it is not defined.

As per the suggestion in the reviewer comment #1, this Figure has now been edited and Panel 5e is now removed.

14. Figure 6a and similar plots: suggest indicating individual subjects with light-color lines.

We have edited Figures 6a and 7a to add individual lines for participants as suggested.

Please note that we decided to not do the same for Figure 8 as this would make the Figure very busy and difficult to read. For the same reason, we also decided to remove the error bars from this figure and restrict the standard error of the mean visualisation to the shaded area around the mean lines.

15. Experiment 4, p. 20 please add a mention of the actual (smaller) number of participants in each group somewhere.

We have added this.

16. Figure 9, suggest note that the behavioral data comes from Exp 2.

We have added this note.

17. Discussion, p. 28, regarding hippocampus and entropy, Bornstein and Daw, 2013, PloS CB would be a good citation to add.

We thank the reviewer for their suggestions. We have now cited the paper.

18. Discussion, end of p. 30, Wimmer et al. 2023 PNAS would be a recent citation in human MEG to add.

We thank the reviewer for their suggestions. We have now cited the paper.

19. Table S3: note reference group, as done in S7.

This has now been noted.

We thank the reviewer for their very helpful comments.

Reviewer #2:

Remarks to the Author:

This manuscript examines the role of the hippocampus in decision making in situations of uncertainty. It proceeds by comparing the performance of patients with autoimmune limbic encephalitis (ALE), a condition that primarily yields lesion of the hippocampus, to that of controls across four experiments featuring decision trials varying in reward amount, required effort, and uncertainty. The hippocampus is a brain region known to be dedicated to memory, and its role in decision making still needs to be elucidated: the results of the current paper contribute nicely to this area of research and should be of interest to the scientific community. Considering the rareness of ALE, the sample size (N=19 ALE patients and N=19 controls) was quite nice. The experiments and statistical analyses were extensive and appeared to be well conducted, and the open availability of the data and code was appreciated. There were some concerns related to being able to infer that the impairments that were revealed could indeed be attributed to a role of the hippocampus in uncertainty, rather than to memory confounds or to other roles of the hippocampus. Detailed comments related to these concerns are provided below for the authors to consider.

(1) Addressing possible memory confounds. One core difficulty in research examining the role of the hippocampus in decision making is ensuring that there is no memory confound, with concern for example that patients may partially forget instructions mid-task and get confused, resulting in impaired decision making. The authors attended to these concerns by making the tasks very visual and keeping some visual cues on the screen during the trials. Nonetheless, the tasks were quite complex, and the full instructions were not displayed on the screen at all times. How did the authors ensure that impairment was not related to confusion or to some memory failure when the tasks became more complex, especially when participants carried out up to 200 decisions in a row without the consequences being played out?

A lack of difference in response reaction time was mentioned as a way to indicate that the trials were not experienced as generally "more difficult" for the patients. But random responding could nonetheless be carried

out quickly. The inclusion of catch trials would have been more effective for checking that patients were still on task.

There was some mention of catch trials in the task related to estimating uncertainty (c.f. caption of Figure 4 and text on page 41). These catch trials were not detailed in the method or results, though, and more detail is warranted to clarify what they consisted of and how the patients fared compared to the controls. But more importantly, were catch trials also included during the decision trials of interest to ensure lack of random responding? And if so, what did they look like and how did the patients fare compare to the controls? No matter what, the topic of a potential memory-based or confusion-based confound should probably be thoroughly discussed in the discussion section as a potential limitation for the present conclusions.

We thank the reviewer for their insightful comment. There are several reasons why, in our opinion, the pattern of the results that we report is not related to memory or general cognitive deficits.

First, active steps were taken to ensure this during and after the experiments, in addition to task design aspects noted in the reviewer's comment.

i) All participants completed a task comprehension quiz after completing training for the task. They had to answer all questions correctly in order to be eligible to perform the task and to be included in the study. We attach this form with this letter and can add it to Supplementary materials.

ii) An experienced researcher (B.A.) was always present to go through the main instructions with the patients. They were also available to answer any questions and intervene if a patient is felt to be confused or randomly responding.

iii) Visual cues were available to reduce memory load as correctly noted by the reviewer. The purpose of these visual cues was also tested in the task comprehension quiz mentioned above.

iv) All participants completed a debriefing questionnaire with the experimenter. This gave us further insights into how people performed the task and the strategies that they used, including if they randomly responded. We now attach this form along with example responses from patients which qualitatively indicate that patients had a good grasp and maintained understanding of the task even after it finished.

Here are some of the answers that the patients gave to the question about Exp. 2: *“During the game, what made you choose whether to answer ‘Yes’ or ‘No?’”*. These answers align with the behavioural results as patients seem to be conscious of the fact that they put more emphasis on uncertainty compared to reward.

Patient 1: *“The probability of getting the answer [circle location] correct”*

Patient 2: *“The likelihood of getting the circle in the right place”*

Patient 3: *“I responded Yes based mostly on the probability of getting a close definition of the circle; this was largely irrespective of the offer, though I did occasionally risk it when the offer was higher”*

Patient 4: *“Based on the size and shape of the purple dots and how close white ones were as well. So spread out purple and close white make it easier to place the circle. I paid almost no attention to the stake/credit altered”*

Patient 5: *“I balanced the best option of finding the circle accurately, against the number of credits to be awarded. I was more invested in placing the circle accurately but was aware the credit score was also an outcome to aim for.”*

Second, analytically, the results also do not support memory confounds

i) There is no significant correlation between cognitive performance as indexed by Adenbrook’s cognitive examination (ACE III) and the main results of the study in ALE patients (Robust regression; ACE III ~ Reward Sensitivity: $R^2 = 0.18$, $T(17) = 0.59$, $p = 0.55$; ACE III ~ Uncertainty Sensitivity: $R^2 = 0.027$, $t(17) = 0.68$, $p = 0.50$). Examination of the correlation between the subdomains of ACE III (Attention, Fluency, Language, Memory, and Visuospatial) also shows no significant correlation with sensitivity to reward or uncertainty (see Table S19 below).

	Reward Sensitivity	Uncertainty Sensitivity
(Intercept)	$\beta = -5.9$ $SE = 7.31$ $t_{13} = -0.81$ $p = 0.43$	$\beta = +10.8$ $SE = 14.1$ $t_{13} = +0.76$ $p = 0.46$
Attention	$\beta = +0.022$ $SE = 0.12$ $t_{13} = +0.18$ $p = 0.86$	$\beta = +0.32$ $SE = 0.233$ $t_{13} = +1.38$ $p = 0.19$
Fluency	$\beta = -0.0834$ $SE = 0.0971$ $t_{13} = -0.86$ $p = 0.41$	$\beta = +0.096$ $SE = 0.188$ $t_{13} = +0.51$ $p = 0.62$
Language	$\beta = +0.0437$ $SE = 0.301$ $t_{13} = +0.15$ $p = 0.89$	$\beta = -0.41$ $SE = 0.584$ $t_{13} = -0.70$ $p = 0.50$
Memory	$\beta = +0.133$ $SE = 0.0737$ $t_{13} = +1.81$ $p = 0.09$	$\beta = +0.0312$ $SE = 0.143$ $t_{13} = +0.22$ $p = 0.83$
VisuoSpatial	$\beta = +0.156$ $SE = 0.496$ $t_{13} = +0.31$ $p = 0.76$	$\beta = -0.337$ $SE = 0.96$ $t_{13} = -0.35$ $p = 0.73$
$adj - R^2$	0.07	0.04
N_{obs}	19	19
AIC	54.02	63.63

Table S19: **Robust regression model investigating the correlation between the subdomains of Addenbrooke's Cognitive Examination (ACE III) and sensitivity to reward and uncertainty in ALE patients. Behavioural data is from Exp. 2.** Models were specified as follows: Sensitivity $\sim 1 + \text{Attention} + \text{Memory} + \text{Fluency} + \text{Language} + \text{VisuoSpatial}$.

ii) If random responding were to happen due to forgetting or confusion one would expect this to be more present towards the end of the experiment compared to the beginning due to decay. Therefore, the performance would be different towards the end of the experiment compared to the start, given that trials run in random order (note that Exp. 4 trials are blocked based on whether uncertainty is present or not). We examined the effect of whether decisions were made in the first half vs. the second half of the experiments. There was no significant interaction between reward and this variable in Exps. 2, 3 & 4 (Reward x Task Half; Exp 2: $\beta = 0.21$, $t(1848) = 1.16$, $p = 0.21$; Exp 3: $\beta = 0.35$, $t(2364) = 1.39$, $p = 0.16$; Exp. 4: $\beta = -0.03$, $t(1848) = -0.24$, $p = 0.8$), going against the assumption that blunted reward sensitivity in ALE is due to forgetting. An in-depth investigation of such inter-trial changes and differences is beyond the scope of this paper and might require different behavioural and analytical approaches. However, for completeness, we have also run a version of the generalised mixed models used to analyse these experiments (Exp.2-4) with per-trial random effect. These models did not reveal any significant changes in the results. We now report these models in Table S20.

iii) In line with the argument made in ii, one would also expect patients to see impaired sensitivity to all decision attributes (reward, effort and uncertainty) with random responding, rather than having a selective impairment pattern sparing many task features and parameters as observed from the results reported. For example, if random responding took place in Exp. 2 in ALE patients, we would have observed blunted sensitivity to both reward and uncertainty, not just reward. Similarly in Exp. 4, ALE patients showed intact uncertainty sensitivity.

iv) We added catch trials in Exp. 4 to both confidence rating and decision phases of the task. We acknowledge that the explanation and analysis of these trials were limited in our original submission. We have now expanded this in the revised manuscript.

In the confidence rating phase, catch trials were added to expand the range of the uncertainty that participants are asked to subjectively estimate which allowed us to better characterise these estimates in a similar way as in Exp. 2 and detect response to sudden changes (Results reported in Figure 10b include these catch trails). These catch trials included 10 trials that could be categorized into high and low levels of uncertainty (five each) outside the uncertainty range that participants would be making decisions against in the decision phase in the 'uncertainty present' condition. The following was added to the Methods section explaining Exp. 4.

"In Exp. 4, a novel version of Circle Quest (Exps. 1 & 2) was designed to investigate effort-based decision making under uncertainty (Figure 4). Participants were familiarised with uncertainty, reward and effort cues using the same training as in Exps. 1–3. This was done in three stages: i) Circle localisation training similar to what was done in Exp. 1 & 2. iii) Confidence rating. Unlike Exp. 2, confidence ratings were blocked and reported during the training phase rather than prior to each decision in the decision phase. This was done to ensure that offers with and without uncertainty were experimentally matched during the decision phase. Ten catch trails were added to increase the range of uncertainty to fully capture subjective estimation of uncertainty as in Exp. 2. These included five trials at the lower range of uncertainty ($EE < 23$ pixels) and five at the higher range ($EE > 91$ pixels) resulting in an uncertainty range of EE between 17.9 and 93.3 pixels. iii) Effort familiarisation and calibration. This was the same as in Exp. 3."

The decision phase included ten catch trials with the same uncertainty levels as the catch trials in the uncertainty estimation phase. They were counterbalanced against the five levels of effort (five trials with low uncertainty and five with high uncertainty) and randomly against the different reward levels. The following was added to the Methods sections:

"Catch trials (10 in total, one in each uncertainty block) were also included in the decision phase. These were the catch trials used in the confidence rating phase that featured different uncertainty levels than what was

otherwise used in the experiment. Each of the two levels of uncertainty (high and low) in these catch trials featured the five levels of effort. Such trials were designed to detect random responding in the task, especially when compared to general task performance.”

The results from these trials further support the argument that memory and confusion effects resulting in random responding have limited effects on performance and the conclusions made in this paper.

There is no significant difference between ALE patients and controls in uncertainty estimation during catch trials (ALE×Uncertainty: $\beta = 0.005$, $t(206) = 0.05$, $p = 0.96$). This is similar to the results reported in the submitted manuscript across all uncertainty estimation trials in Exp. 2 and Exp .4 (Figure 10).

ALE patients also demonstrated intact sensitivity to uncertainty in decision catch trials (ALE×Uncertainty: $\beta = -0.91$, $t(201) = -0.20$, $p = 0.13$) and were less sensitive to effort than controls (ALE×Effort: $\beta = 6.63$, $t(201) = 2.74$, $p < 0.01$). These results again show that ALE decision performance is not likely to be random or driven by a lack of understanding of the task. We found no significant difference between ALE patients’ and controls’ reward sensitivity in catch trials ($\beta = -2.67$, $t(201) = -1.65$, $p = 0.15$), however, the interpretation of this result should be limited as reward was not properly counterbalanced in these trial. It would require 40 catch trials to fully capture the response to all three variables equally.

We summarised these analyses and results in Results section in the revised manuscript:

“In addition to these control analyses, we investigated the possible effect of cognitive deficit and memory decay on performance. There was no significant correlation between cognitive performance indexed by ACE III scores (whether total or subdomains) and sensitivity to either reward or uncertainty in Exp. 2 (Robust regression: ACE III ~ Reward Sensitivity in Exp. 2: $R^2 = 0.18$, $t_{17} = 0.59$, $p = 0.55$; ACE III ~ Uncertainty Sensitivity: $R^2 = 0.028$, $t_{17} = 0.68$, $p = 0.50$; Figure S8a., Table S19).

To investigate whether performance (mainly reward sensitivity) was influenced by task duration across the different tasks in ALE we analysed decisions in the first half compared to the second half in Exps. 2–4. This showed that reward sensitivity across the different tasks did not differ between the two task halves in ALE patients, indicating minimal effect of memory decay during task performance (Reward×2nd Task Half; Exp. 2: $\beta = 0.21$, $t_{1848} = 0.25$, $p = 0.25$; Exp. 3: $\beta = 0.35$, $t_{12364} = 1.39$, $p = 0.16$; Exp. 3: $\beta = -0.039$, $t_{1592} = -0.25$, $p = 0.80$, Figure S8b., Table S21).

Finally, we analysed the catch trials from Exp. 4 to investigate features that might suggest random responding (e.g., sensitivity to sudden changes in uncertainty levels). The results from these trials showed that ALE patients responded as expected with intact sensitivity to sudden uncertainty changes both during the uncertainty estimation phase (ALE×Uncertainty: $\beta = 0.0056$, $t_{206} = 0.05$, $p = 0.96$) and the decision phase (ALE×Uncertainty: $\beta = -0.914$, $t_{201} = -0.20$, $p = 0.84$, Figure S8c., Table S22). ALE patients also showed blunted sensitivity to effort during these trials as demonstrated in the rest of the experiment (ALE×Effort = $\beta = 6.63$, $t_{201} = 2.74$, $p < 0.01$). Analysis of reward responsiveness in these trials is restricted because of their unbalanced design that does not feature all reward levels equally, in addition to the small sample size. “

Figure S7: Minimal effect of cognitive deficit and memory decay on performance. **a.** There was no significant correlation between cognitive scores indexed by ACE-III scores and sensitivity to either reward or uncertainty in Exp. 2, indicating that the difference between ALE patients and controls is likely related to cognitive dysfunction. **b.** Whether trials were played in the second half of the experiment compared to the first half did not have a significant effect on reward sensitivity, suggesting minimal presence of memory decay that could influence behaviour or result in random responding. **c.** Catch trials in Exp. 4 show that ALE patients had intact sensitivity to uncertainty (right panel) and blunted sensitivity to effort (middle panel), pointing against random responding during the task, and replicating results from the main task trials. Reward sensitivity (left panel) is intact in these trials but this should be interpreted with caution as reward represented in a balanced design in these catch trials. Line shadows indicate \pm SEM. For statistical details see Tables S18, S20 S21.

Finally, we have added a paragraph discussing the issue of memory decay and confusion confounds and how they might limit the interpretation of the results. Note that we also highlight the mental-time travel dysfunction and its effect on decision making could also be conceptualised using uncertainty-centred view (see also our reply to your comment #2.a).

“It is, however, challenging to fully ascertain whether the results of this study reflect a specific computational property of the hippocampus or instead, a general disruption of cognitive processing that might be observed with other brain lesions. Three factors make this possibility unlikely: i) The correlation between behaviour and severity of hippocampal atrophy, rather than other closely related regions such as the amygdala, which might have been affected by the disease process as well (see Figures 9 & S5). ii) The results do not correlate with cognitive dysfunction indexed by ACE-III scores (both total and subdomains, including memory) or meta-cognitive deficits in uncertainty estimation (Figure S8a., Tables S19 & S18). iii) The analysis of additional experimental parameters, including performance on catch trials and closer examination of decisions made around task onset and finish points (Figure S8b-c.), contradicts the idea that performance is a reflection of cognitive dysfunction or random responding. This is especially notable considering that the main results exhibit a selective deficit not globally affecting all value attributes as one would expect with chance-level performance. Moreover, the task design and administration have been tailored to minimise such effects including, completing comprehension and debriefing questionnaires, using interactive tutorials for training, and adding cues to reduce memory load. Nevertheless, despite these considerations, it might be impossible to completely rule out the effect of cognitive or memory deficits on decision-making in hippocampal patients, and this could be a potential limitation of this study. It would be more reasonable to aim to interpret the findings within the broader context of hippocampal episodic and memory functions, rather than in isolation, as highlighted in our previous discussions (e.g., considering similarities between inter-temporal decisions and uncertainty).”

(2) Implications/Discussion of findings. Some of the implications of the present findings and comparison with the literature could be strengthened in the Discussion section.

2.a. First, I am not sure that I followed the direct link that was made between episodic mental projection of future events and intolerance of uncertainty (pages 27-28). These sections could benefit from clarifications.

We have added more clarification in the Discussion section:

"In a broader conceptual context, such scenarios necessitating value inference can be regarded as forms of decision making under uncertainty. For instance, examining delay discounting in inter-temporal decisions reveals cognitive and computational parallels with probabilistic discounting that characterises uncertainty valuation (Green and Myerson, 1996; Myerson et al., 2003; Prelec and Loewenstein, 1991; Rachlin et al., 1991; Stevenson, 1986). This resemblance might stem from the inherent risk associated with both discounting properties, representing the likelihood of obtaining probabilistic or delayed rewards (Myerson et al., 2003; Stevenson, 1986). But also it might reflect a temporal aspect in probabilistic discounting, where agents consider the attempts required to secure certain probabilistic rewards and the time investment (Rachlin et al., 1991). Notably, the hippocampus assumes a pivotal role in inter-temporal decision processes, particularly when they involve episodic future thinking, where the inclusion episodic details of future rewards reduces delay discounting (Palombo et al., 2015b; Peters and Buchel, 2010; Ye et al., 2021). Considering these perspectives in conjunction with findings from our study, an alternative interpretation centred on uncertainty emerges. For example, exaggerated delay seen in patients with hippocampus damage in previous reports might signify increased uncertainty of future rewards.

Such reports highlighting a potential role of the hippocampus in uncertainty processing are supported by functional neuroimaging studies which have demonstrated hippocampal activation that correlates with the degree of uncertainty (entropy) of sensory stimuli when making decisions (Bornstein and Daw, 2013; Harrison et al., 2006; Rigoli et al., 2019; Strange et al., 2005; Tobia et al., 2012). These investigations align with the view that regards uncertainty as a threatening stimulus (i.e., carries risk signals) processed by hippocampus-centred behavioural inhibition system (BIS) (Gray and McNaughton, 2003). The hippocampus, according to this view, is considered to work as a mismatch detection system comparing expectation with perceived stimuli and triggering behavioural avoidance when confronted by uncertainty (or other anxiety-inducing stimuli) (Gray and McNaughton, 2003)."

2.b. It was also not clear to me that findings of impairments when reward & uncertainty demands were present but not when reward & effort demands were present do not simply reflect that the task with uncertainty demands was more complex than the task with effort demands. Reaction time was compared across participant groups. But was reaction time compared across tasks?

We have now compared reaction times across three tasks. The informative comparison is between Exp. 2 & Exp. 3 given that they both feature only two attributes (compared to three attributes in Exp. 4) and have the same number of participants. There was no significant difference in reaction time between Exp. 2 & Exp. 3 across all participants ($\beta = 0.128$, $t(91) = 0.94$, $p = 0.35$) and the interaction of groupXExp was also not significant ($\beta = 0.225$, $t(91) = 1.17$, $p = 0.25$). Comparing reaction time within the ALE group across the two Exps.

reveals that ALE patients actually take longer in Exp. 3 (reward & effort) compared to Exp. 2 (reward & uncertainty) (Exp. 2: Mean = 2.04, SD = 0.40, Exp 3: Mean = 2.40, SD = 0.49, $t(18) = 2.25$, $p = 0.03$). This could indicate that trials in Exp. 2 require less deliberation for ALE patients, perhaps consistent with disregarding other values (reward) in the presence of uncertainty, which is evident in their preferences as well as in active samples (Exp. 1) featuring faster sampling rates.

Also note that, as expected, Exp. 4 has significantly increased decision time ($\beta = 1.18$, $t(91) = 7.68$, $p < 0.0001$), given the more difficult decision requiring the consideration of three attributes. This result also provides another evidence against fast rapid responding discussed in our reply to the first comment.

While such findings might provide insights into how difficult task performance is, we agree that reaction times do not entirely rule out the presence of a complexity effect. It might be necessary to conduct novel experiments with a redesigned task, requiring participants to infer uncertainty and effort levels using analogous cues (e.g., levels on a bar). This approach aims to eliminate any additional cognitive effort needed to infer these attributes.

We have briefly added one result to Exp. 3 Results section as follows:

“Compared to Exp. 2, these decision times were slower (Exp. 2: $\mu = 2.04$, $SD = \pm 0.40$, Exp 3: $\mu = 2.40$, $SD = \pm 0.49$, $t(18) = 2.25$, $p = 0.03$), indicating less deliberation when making decisions under uncertainty in Exp.2 compared to effort-based decisions (Figure S2, Table S23, see Supplementary materials for extended results analysing decision times across Exp. 2–4).”

We have also provided the details of these results in the Supplementary material sections along with Figure S2 and Table S22 shown below.

Figure S2. Decision times measured in seconds (sec) in Exps. 2–4. Across passive decision tasks (Exps. 2–4), no significant difference was found between ALE patients and controls. ALE patients made faster decisions in Exp. 2 compared to Exp. 3, indicating less deliberation when making decisions under uncertainty compared to effort-based decision making. Exp. 4 had significantly slower decisions, reflecting the more complex task structure with three decision attributes to consider. For full statistical details see Table S23.

Decision Time Exps. 2–4	
(Intercept)	$\beta = +2.1$ $SE = 0.116$ $t_{91} = +18.22$ $p < 0.0001$
Exp.3	$\beta = +0.128$ $SE = 0.136$ $t_{91} = +0.94$ $p = 0.35$
Exp.3:ALE	$\beta = +0.225$ $SE = 0.192$ $t_{91} = +1.17$ $p = 0.25$
Exp.4	$\beta = +1.18$ $SE = 0.154$ $t_{91} = +7.68$ $p < 0.0001$
Exp.4:ALE	$\beta = -0.311$ $SE = 0.241$ $t_{91} = -1.29$ $p = 0.20$
ALE	$\beta = -0.057$ $SE = 0.163$ $t_{91} = -0.35$ $p = 0.73$
$adj - R^2$	0.60
N_{obs}	97
AIC	151.12

Table S23: **Decision Time across Exps. 2–3.** Models were specified as follows. Decision Time: $DT \sim 1 + Exp*group + (1 | Participant)$. Exp. 2 was set as the reference Exp.

The discussion on this issue was expanded with further acknowledgement of this limitation.

“ In a similar vein, it could be argued that the results might merely reflect a difficulty effect imposed by more complex demands of uncertainty cues compared to other cues. However, if this were the case, one would expect patients to take longer than controls when making decisions under uncertainty (Exp. 2). Contrarily, this expectation did not align with the observed results. It would also be predicted to find slower reaction times in Exp. 2 (involving reward and uncertainty) compared to Exp. 3 (involving reward and effort). On the contrary, the results demonstrated the opposite trend, indicating shorter deliberation time in ALE patients when confronted with uncertainty. This performance is consistent with the decision-making pattern observed in the study that tends to disregard other attributes in the presence of uncertainty, potentially leading to quicker decisions centred around uncertainty. A similar tendency is also evident in the active sampling experiment (Exp. 2), displaying faster sampling rates among ALE patients, compared to controls. While such analyses indirectly provide some insights into complexity effects on performance, future studies might want to focus on disentangling this experimentally in task designs that feature analogous cues for the attributes being measured. Along these lines, it would also be insightful to contextualise and establish the empirical connection between the observed pattern of the results in this study and other broader spectrum of potential hippocampal roles in goal-directed behaviour such as task representation from integrating complex features (Mizrak et al., 2021; Samborska et al., 2022; Yonelinas, 2013) and experimental range adaptation (Cox and Kable, 2014). Impairment in these functions might lead to diminished sensitivity to some task features when agents are required to integrate them to guide decisions and behaviour (Bett et al., 2015; Masuda et al., 2020; Patt et al., 2023).”

2.c. The present results are interestingly quite consistent with previous studies that have reported an impairment in sensitivity to change in one task feature when decisions required integrating or updating several task components. For example, in the intertemporal choice animal hippocampal lesion literature, Bett et al. 2015 (Hippocampus, DOI 10.1002/hipo.22400) demonstrated diminished sensitivity to change in the rewards, and Masuda et al. 2020 (eLife, DOI: <https://doi.org/10.7554/eLife.52466>) demonstrated diminished sensitivity to changes in the delay. Similarly, in a more recent human study on patients with hippocampal lesions, Patt et al. 2023 (Journal of Neuroscience, DOI: <https://doi.org/10.1523/JNEUROSCI.2250-22.2023>) demonstrated diminished sensitivity to changes in the delay during experiential intertemporal choice.

These similar patterns of findings could be explained by a role of the hippocampus in processes other than uncertainty, for example from the role of the hippocampus in constructing representations that require integrating several complex task features (Yonelinas 2013, Beh Brain Research, 254, p.34) or from a role of the hippocampus in experimental range adaptation (Cox & Kable, 2014, 34, p.16533 J. Neuroscience). Some of these considerations could be added in the Discussion section.

We thank the reviewer for their insightful comment. Some of these considerations are added to the Discussion section.

“Along these lines, it would also be insightful to contextualise and establish the empirical connection between the observed pattern of the results in this study and other broader spectrum of potential hippocampal roles in goal-directed behaviour such as task representation from integrating complex features (Mizrak et al., 2021; Samborska et al., 2022; Yonelinas, 2013) and experimental range adaptation Cox and Kable (2014). Impairment in these functions might lead to diminished sensitivity to some task features when agents are required to integrate them to guide decisions and behaviour (Bett et al., 2015; Masuda et al., 2020; Patt et al., 2023)”

Some of these considerations are also discussed in our discussion of Inter-temporal decision making and uncertainty in a different section (see our reply to your comment #2.a).

We thank the reviewer for very helpful comments.

Reviewer #3:

Remarks to the Author:

In their manuscript entitled “Role of the hippocampus in decision making under uncertainty” Attaallah et al. hypothesize that the human hippocampus may play a specific role in decision making involving evaluation of uncertain values.

They studied a group of 19 individuals with acute autoimmune limbic encephalitis with anti-LGI1 and/or CASPR2 antibodies deemed to focally affect the hippocampus compared to 19 healthy age- and gender-matched controls on how they evaluate reward against uncertainty compared to reward against physical effort using four experiments ((i) active information gathering prior to committing to decisions under uncertainty using the active Circle Quest paradigm, (ii) passive decision making under uncertainty using the passive Circle Quest paradigm, (iii) effort-based decision making, and (iv) effort-based decision making under uncertainty) requiring participants to make trade-offs between reward, uncertainty and effort.

They obtained the following major findings:

1. ALE patients demonstrated blunted sensitivity to reward and effort whenever uncertainty was considered, despite demonstrating intact uncertainty sensitivity.
2. By contrast, valuation of reward and effort was intact on uncertainty-free tasks.
3. Reduced sensitivity to changes in reward under uncertainty correlated with severity of hippocampal damage.

Authors conclude a context-sensitive role of the hippocampus in value-based decision making specifically under conditions of uncertainty.

This is a very well designed and conducted study demonstrating for the role of the hippocampus in decision making and extend our knowledge regarding cognitive/behavioural deficits in ALE patients.

There are some minor issues that should be addressed by the authors:

Clinical details regarding the ALE patients are warranted. Especially, disease duration and treatments need to be described.

Some of the clinical details have been included in the submitted manuscripts in Table S2 (Patients Characteristics) including disease duration (Years Since Diagnosis) and antibody status (Abs). We have now extended the clinical details. A new column (Years Since First Symptoms) was added to S2 to indicate the time between the first reported symptoms and assessment in our experiments. An additional Table S3 has now been added to provide clinical details including, symptom profiles in the acute and chronic phase as well as treatments in both phases.

Code	Age	Gender	Abs	Lt. Hipp.		Rt. Hipp.		Years Since Diagnosis	Years Since First Symptom
				Raw Volume (adjusted)	Percentile	Raw Volume (adjusted)	Percentile		
1	53	F	LGII	3559.64 (3710.42)	29	3547.33 (3725.45)	21	3.38	5.27
2	47	F	LGII	3157.67 (3111.00)	6 ^c	2717.64 (2662.51)	<2.5 ^c	3.44	3.77
3	59	F	LGII	2332.83 (2313.53)	<2.5	2255.53 (2232.73)	<2.5	3.49	9.74
4	63	F	LGII	2860.65 (2941.60)	<2.5	3744.13 (3839.76)	43	2.59	2.92
5	72	F	LGII	3170.72 (3204.72)	17 ^c	2495.43 (2535.59)	<2.5 ^c	7.67	7.67
6	64	M	LGII	-	-	-	-	4.66	4.82
7	55	M	LGII	4835.20 (4969.31)	97	4125.12 (4283.55)	46	2.56	2.64
8	53	M	LGII	3663.60 (3710.05)	19	3754.24 (3809.11)	17	2.36	3.21
9	66	M	LGII	4109.11 (4115.25)	72	4341.71 (4348.97)	80	1.18	1.77
10	65	M	LGII	3488.01 (3240.14)	18	3379.20 (3086.38)	9	3.27	4.11
11	72	M	LGII	2973.83 (2811.09)	5	2905.04 (2712.78)	3	1.82	1.9
12	26	M	LGII	-	-	-	-	0.96	0.97
13	68	M	CASPR2	3050.27 (2970.48)	4	3364.09 (3269.84)	10	1.03	1.08
14	77	M	CASPR2	-	-	-	-	10.52	10.52
15	65	M	CASPR2	3455.68 (3311.09)	16	3411.77 (3240.95)	10	5.29	6.87
16	67	M	CASPR2	4118.84 (4163.74)	74	3942.82 (3995.87)	48	4.30	5.47
17	58	F	LGII/CASPR2	3670.25 (3590.67)	41	3073.41 (2979.41)	4	3.26	3.83
18	58	M	LGII/CASPR2	-	-	-	-	7.16	7.16
19	52	M	Seronegative	2370.64 (2228.48)	<2.5 ^c	2752.98 (2585.05)	<2.5 ^c	1.99	2.78

Table S2: **Patients Characteristics.** Abs: Autoantibodies. Lt. Hipp.: Left Hippocampus. Rt. Hipp.: Right Hippocampus. Hippocampal volumes were adjusted for intra-cranial volumes. Percentile is determined by plotting raw hippocampal volumes against normative brain volumes from UK biobank data (Nobis et al., 2019). ^c: Describes percentiles outside the age range of the UK biobank nomograms. Percentiles according to the closest age value within the UK biobank range was used instead.

Code	Clinical Profile on Presentation	Clinical profile in chronic phase	Acute management	Medications in chronic phase
1	Memory deficits, irritability, falls	Seizures	IVIG (x3), PLEX (x1), Steroids	Steroids (low dose), Levetiracetam, Lamotrigine
2	Seizures (numbness and weakness left hand), anxiety, fatigue	Memory deficits, emotional liability, abnormal sensation left hand	na	Steroids (low dose), Mycophenolate, Carbamazepine
3	Seizures, memory deficits	Seizures	Steroids, azathioprine (briefly), methotrexate, carbamazepine, lacosamide, PLEX	Carbamazepine, Lacosamide
4	FBDS, memory deficits, auditory hallucinations, anxiety, falls	Poor concentration, fatigue, apathy	Steroids, Azathioprine, PLEX, levetiracetam, mycophenolate, Rituximab	Steroids (low dose), Mycophenolate, Levetiracetam
5	Neurocardiac syndrome (tachi-bradycardia), seizures (thermal and sensory sensations and one tonic-clonic), increased daytime sleepiness, headache, fatigue, brain fog	Anxiety, abdominal sensations	Steroids, levetiracetam	Levetiracetam
6	Behavioural changes, nocturnal seizures	No symptoms	na	Carbamazepine
7	FBDS, memory deficits	Memory deficits	Steroids, Clobazam, PLEX	Steroids (low dose), Pregabalin
8	Seizures (including hysterical laughing), hypersomnia, anxiety, memory deficits, cough, breathlessness	No symptoms	IVIG, Steroids, lacosamide	Steroid (low dose), Mycophenolate, Lacosamide
9	FBDS	No symptoms	Steroids, Lamotrigine	Steroids (lower dose), Lamotrigine
10	Seizures, behavioural change (apathy), memory deficits	Apathy, memory deficits	na	None
11	Seizures (tingling, lateralized weakness), memory deficits	Memory deficits	Steroids, Lamotrigine	Steroids (low dose), Lamotrigine
12	FBDS, memory deficits	No symptoms	Steroids, PLEX, Levetiracetam	Steroids (low dose), Levetiracetam
13	Seizures, memory deficits, emotional liability, sleep cycle inversion	Memory and concentration deficits, headaches, leg pain	Steroids, levetiracetam, PLEX, Pregabalin	Steroids (low dose), Levetiracetam, Pregabalin
14	Morvan's syndrome, speech and balance problems	Neuropathic pain, problems with balance, muscle twitching, memory deficits	Mycophenolate, Steroids, Pregabalin	Steroids (low dose), Pregabalin, Sertraline
15	Seizures, memory deficits	Memory deficits	Lamotrigine, Steroids	Lamotrigine
16	Seizures, behavioural change, hallucinations	Memory deficits, fatigue	Lacosamide	Lacosamide
17	Lower limbs pain, insomnia, muscle twitching, sweating, abdominal bloating, Morvan's syndrome, rash, confusion, hallucinations, seizures	Pain and numbness in lower limbs, muscle twitching, fatigue	Steroids, IVIG, PLEX, Cyclophosphamide, Phenytoin, pPregabalin, Mirtazapine	Steroids (low dose), Phenytoin, Pregabalin
18	Memory deficits	Memory deficits	Cyclophosphamide, Phenytoin	Steroids (low dose), Levetiracetam
19	Seizure, fatigue, apathy, delusion, irritability	Seizures, Memory deficits, anxiety, verbally aggressive	Steroids, Lacosamide, Levetiracetam, Citalopram	Lacosamide, Levetiracetam, Citalopram

Table S3: **Patients Clinical Profiles.** FBDS: Faciobrachial dystonic seizures. IVIG: intravenous immunoglobulin. PLEX: Plasma exchange. na: data not available from patients local records.

How were ALE and HC groups matched regarding education?

Participants were not matched regarding education. This is mainly due to two reasons: ii) no effect of education or intelligence scores (indexed by Raven Matrices score) on behavioural performance during piloting, validation and testing in our previous studies using the same tasks (Petitet et al., 2021 Nat. Hum Beh., Attaallah et al., 2022, eLife). ii) Difficulty recruiting healthy controls that match the patients' demographics, especially during the pandemic. Matching for education would have posed more restrictions so we opted for age- and gender

matching. However, we acknowledge this concern which also extends to confusion, difficulty and cognitive deficits effects that might have influenced behavioural performance. We have now extensively addressed these issues in our revised manuscript with new analyses and discussions.

“In addition to these control analyses, we investigated the possible effect of cognitive deficit and memory decay on performance. There was no significant correlation between cognitive performance indexed by ACE III scores (whether total or subdomains) and sensitivity to either reward or uncertainty in Exp. 2 (Robust regression: ACE III ~ Reward Sensitivity in Exp. 2: $R^2 = 0.18$, $t_{17} = 0.59$, $p = 0.55$; ACE III ~ Uncertainty Sensitivity: $R^2 = 0.028$, $t_{17} = 0.68$, $p = 0.50$; Figure S8a., Table S19).

To investigate whether performance (mainly reward sensitivity) was influenced by task duration across the different tasks in ALE we analysed decisions in the first half compared to the second half in Exps. 2–4. This showed that reward sensitivity across the different tasks did not differ between the two task halves in ALE patients, indicating minimal effect of memory decay during task performance (Reward×2nd Task Half; Exp. 2: $\beta = 0.21$, $t_{1848} = 0.25$, $p = 0.25$; Exp. 3: $\beta = 0.35$, $t_{12364} = 1.39$, $p = 0.16$; Exp. 3: $\beta = -0.039$, $t_{1592} = -0.25$, $p = 0.80$, Figure S8b., Table S21).

Finally, we analysed the catch trials from Exp. 4 to investigate features that might suggest random responding (e.g., sensitivity to sudden changes in uncertainty levels). The results from these trials showed that ALE patients responded as expected with intact sensitivity to sudden uncertainty changes both during the uncertainty estimation phase (ALE×Uncertainty: $\beta = 0.0056$, $t_{206} = 0.05$, $p = 0.96$) and the decision phase (ALE×Uncertainty: $\beta = -0.914$, $t_{201} = -0.20$, $p = 0.84$, Figure S8c., Table S22). ALE patients also showed blunted sensitivity to effort during these trials as demonstrated in the rest of the experiment (ALE×Effort = $\beta = 6.63$, $t_{201} = 2.74$, $p < 0.01$). Analysis of reward responsiveness in these trials is restricted because of their unbalanced design that does not feature all reward levels equally, in addition to the small sample size. “

Figure S7: Minimal effect of cognitive deficit and memory decay on performance. **a.** There was no significant correlation between cognitive scores indexed by ACE-III scores and sensitivity to either reward or uncertainty in Exp. 2, indicating that the difference between ALE patients and controls is likely related to cognitive dysfunction. **b.** Whether trials were played in the second half of the experiment compared to the first half did not have a significant effect on reward sensitivity, suggesting minimal presence of memory decay that could influence behaviour or result in random responding. **c.** Catch trials in Exp. 4 show that ALE patients had intact sensitivity to uncertainty (right panel) and blunted sensitivity to effort (middle panel), pointing against random responding during the task, and replicating results from the main task trials. Reward sensitivity (left panel) is intact in these trials but this should be interpreted with caution as reward represented in a balanced design in these catch trials. Line shadows indicate \pm SEM. For statistical details see Tables S18, S20 S21.

We added an extended analysis of reaction times to Supplementary materials to provide insights into difficulty effects:

“Decision times Decision times across the three passive decision making tasks (Exps 2–3) were compared to gain further insights into the cognitive process involved. The informative comparison is mainly between Exp. 2 and Exp. 3 given that they both feature only two attributes (compared to three attributes in Exp. 4) and have an equal number of participants. No significant difference in reaction time emerged between Exp. 2 and Exp. 3 across all participants ($\beta = 0.128$, $t_{91} = 0.94$, $p = 0.35$, Figure S2 Table S23), and the interaction of group \times Exp was also not significant ($\beta = 0.225$, $t_{91} = 1.17$, $p = 0.25$). Comparing reaction time within the ALE group across the two Exps. reveals that ALE patients actually took longer in Exp. 3 (reward & effort) compared to Exp. 2 (reward & uncertainty) (Exp. 2: $\mu = 2.04$, $SD = \pm 0.40$, Exp 3: $\mu = 2.40$, $SD = \pm 0.49$, $t_{18} = 2.25$, $p = 0.03$). This could indicate that trials in Exp. 2 require less deliberation for ALE patients, possibly implying the disregard of other values (such as reward) in the presence of uncertainty. This observation aligns with their preferences and active samples (Exp. 1), which exhibit faster sampling rates. It is noteworthy that Exp. 4, as expected, demonstrated significantly increased decision time ($\beta = 1.18$, $t_{91} = 7.68$, $p < 0.0001$), consistent with the more complex decision-making process involving the consideration of three attributes. This result further supports the argument against rapid responding discussed in the subsection of the Results. While these findings shed light on task performance difficulty, it is important to acknowledge that reaction times may not entirely negate the presence of a complexity effect. To control for this effect, it might be necessary to conduct novel experiments with a redesigned task, requiring participants to infer uncertainty and effort levels using analogous cues (e.g., levels on a bar). This approach aims to eliminate any additional cognitive effort needed to infer these attributes.”

We have added paragraphs discussing the issue of cognitive deficits and difficulty confounds and how they might limit the interpretation of the results.

“It is, however, challenging to fully ascertain whether the results of this study reflect a specific computational property of the hippocampus or instead, a general disruption of cognitive processing that might be observed with other brain lesions. Three factors make this possibility unlikely: i) The correlation between behaviour and severity of hippocampal atrophy, rather than other closely related regions such as the amygdala, which might have been affected by the disease process as well (see Figures 9 & S5). ii) The results do not correlate with cognitive dysfunction indexed by ACE-III scores (both total and subdomains, including memory) or meta-cognitive deficits in uncertainty estimation (Figure S8a., Tables S19 & S18). iii) The analysis of additional experimental parameters, including performance on catch trials and closer examination of decisions made around task onset and finish points (Figure S8b-c.), contradicts the idea that performance is a reflection of cognitive dysfunction or random responding. This is especially notable considering that the main results exhibit a selective deficit not globally affecting all value attributes as one would expect with chance-level

performance. Moreover, the task design and administration have been tailored to minimise such effects including, completing comprehension and debriefing questionnaires, using interactive tutorials for training, and adding cues to reduce memory load. Nevertheless, despite these considerations, it might be impossible to completely rule out the effect of cognitive or memory deficits on decision-making in hippocampal patients, and this could be a potential limitation of this study. It would be more reasonable to aim to interpret the findings within the broader context of hippocampal episodic and memory functions, rather than in isolation, as highlighted in our previous discussions (e.g., considering similarities between inter-temporal decisions and uncertainty)."

" In a similar vein, it could be argued that the results might merely reflect a difficulty effect imposed by more complex demands of uncertainty cues compared to other cues. However, if this were the case, one would expect patients to take longer than controls when making decisions under uncertainty (Exp. 2). Contrarily, this expectation did not align with the observed results. It would also be predicted to find slower reaction times in Exp. 2 (involving reward and uncertainty) compared to Exp. 3 (involving reward and effort). On the contrary, the results demonstrated the opposite trend, indicating shorter deliberation time in ALE patients when confronted with uncertainty. This performance is consistent with the decision-making pattern observed in the study that tends to disregard other attributes in the presence of uncertainty, potentially leading to quicker decisions centred around uncertainty. A similar tendency is also evident in the active sampling experiment (Exp. 2), displaying faster sampling rates among ALE patients, compared to controls. While such analyses indirectly provide some insights into complexity effects on performance, future studies might want to focus on disentangling this experimentally in task designs that feature analogous cues for the attributes being measured. Along these lines, it would also be insightful to contextualise and establish the empirical connection between the observed pattern of the results in this study and other broader spectrum of potential hippocampal roles in goal-directed behaviour such as task representation from integrating complex features (Mizrak et al., 2021; Samborska et al., 2022; Yonelinas, 2013) and experimental range adaptation (Cox and Kable, 2014). Impairment in these functions might lead to diminished sensitivity to some task features when agents are required to integrate them to guide decisions and behaviour (Bett et al., 2015; Masuda et al., 2020; Patt et al., 2023)."

Figure S2. Decision times measured in seconds (sec) in Exps. 2-4. Across passive decision tasks (Exps. 2-4), no significant difference was found between ALE patients and controls. ALE patients made faster decisions in Exp. 2 compared to Exp. 3, indicating less deliberation when making decisions under uncertainty compared to effort-based decision making. Exp. 4 had significantly slower decisions, reflecting the more complex task structure with three decision attributes to consider. For full statistical details see Table S23.

What were the time intervals between behavioural testing and MRI in both groups?

The average time interval between MRI and behavioural testing for the ALE group (N=15) was 49.73 days (SD = 59.00 days). This was shorter than the interval for controls (N=17) which was 109.82 days (SD = 93.93 days) ($z = 2.11$, $p = 0.034$). These intervals are now reported in the revised manuscript under Magnetic Resonance data acquisition subsection of Materials and Methods.

Note that the interval difference between the two groups should have no influence on the results because i) we primarily examine structural MRI features which are mostly stable over these relatively short time intervals and ii) the correlations between MRI hippocampal volumes and behavioural measures are carried out *within* the ALE group.

Why did the authors use absolute values for the hippocampal volume instead of normalizing them to the brain/head volume?

We have actually used adjusted hippocampal volumes for intracranial volume. This was mentioned and detailed under the Magnetic Resonance data analysis subsection under Materials and Methods. Perhaps the confusion might have arisen from the reported raw values in Table S2. We used raw hippocampal volumes to acquire the percentiles of the hippocampal volumes on the normogram from *Nobis et al., 2019*. This is the only analysis where raw volumes were used, as it is a requirement for using the tool which then adjusts volumes and percentiles automatically based on extracted intracranial index parameters acquired with the analysis. We now report both raw and adjusted values in Table S2.

We thank the reviewer for their very helpful comments.

Decision Letter, first revision:

26th January 2024

Dear Dr. Attaallah,

Thank you for your patience as we've prepared the guidelines for final submission of your Nature Human Behaviour manuscript, "Role of the hippocampus in decision making under uncertainty" (NATHUMBEHAV-23082523A). Please carefully follow the step-by-step instructions provided in the attached file, and add a response in each row of the table to indicate the changes that you have made. Please also check and comment on any additional marked-up edits we have proposed within the text. Ensuring that each point is addressed will help to ensure that your revised manuscript can be swiftly handed over to our production team.

We would hope to receive your revised paper, with all of the requested files and forms within two-three weeks. Please get in contact with us if you anticipate delays.

Nature Human Behaviour offers a Transparent Peer Review option for new original research manuscripts submitted after December 1st, 2019. As part of this initiative, we encourage our authors to support increased transparency into the peer review process by agreeing to have the reviewer comments, author rebuttal letters, and editorial decision letters published as a Supplementary item. When you submit your final files please clearly state in your cover letter whether or not you would like to participate in this initiative. Please note that failure to state your preference will result in delays in accepting your manuscript for publication.

In recognition of the time and expertise our reviewers provide to Nature Human Behaviour's editorial process, we would like to formally acknowledge their contribution to the external peer review of your manuscript entitled "Role of the hippocampus in decision making under uncertainty". For those reviewers who give their assent, we will be publishing their names alongside the published article.

Cover suggestions

We welcome submissions of artwork for consideration for our cover. For more information, please see our guide for cover artwork.

ORCID

Non-corresponding authors do not have to link their ORCIDs but are encouraged to do so. Please note that it will not be possible to add/modify ORCIDs at proof. Thus, please let your co-authors know that if they wish to have their ORCID added to the paper they must follow the procedure described in the following link prior to acceptance:

Nature Human Behaviour has now transitioned to a unified Rights Collection system which will allow our Author Services team to quickly and easily collect the rights and permissions required to publish your work. Approximately 10 days after your paper is formally accepted, you will receive an email in providing you with a link to complete the grant of rights. If your paper is eligible for Open Access, our Author Services team will also be in touch regarding any additional information that may be required to arrange payment for your article.

Please note that *Nature Human Behaviour* is a Transformative Journal (TJ). Authors may publish their research with us through the traditional subscription access route or make their paper immediately open access through payment of an article-processing charge (APC). Authors will not be required to make a final decision about access to their article until it has been accepted. Find out more about Transformative Journals

[REDACTED]

Best regards,
Alex McKay

Editorial Assistant
Nature Human Behaviour

On behalf of

Giacomo Ariani
Editor
Nature Human Behaviour

Reviewer #1:

Remarks to the Author:

The revised manuscript has addressed all of my concerns and comments from the initial submission. Congratulations on a very interesting paper.

Reviewer #2:

Remarks to the Author:

I was impressed with the quality of the responses and modifications made to the manuscript in response to my comments. The authors have addressed all my concerns.

Reviewer #3:

Remarks to the Author:

The authors addressed all issues raised by this reviewer.

Final Decision Letter:

Dear Dr Attaallah,

We are pleased to inform you that your Article "The role of the human hippocampus in decision making under uncertainty", has now been accepted for publication in Nature Human Behaviour.

Please note that *Nature Human Behaviour* is a Transformative Journal (TJ). Authors may publish their research with us through the traditional subscription access route or make their paper immediately open access through payment of an article-processing charge (APC). Authors will not be required to make a final decision about access to their article until it has been accepted. Find out more about Transformative Journals

With best regards,

Giacomo Ariani
Editor
Nature Human Behaviour